# A mechanism with severing near barbed ends and annealing explains structure and dynamics of dendritic actin networks

**Danielle Holz[1], Aaron R Hall[1], Eiji Usukura[2], Sawako Yamashiro[2], Naoki Watanabe[2], Dimitrios Vavylonis[1]\***

[1]Department of Physics, Lehigh University, Bethlehem, United States; [2]Laboratory of Single-Molecule Cell Biology, Kyoto University, Kyoto, Japan

**Abstract** Single molecule imaging has shown that part of actin disassembles within a few seconds after incorporation into the dendritic filament network in lamellipodia, suggestive of frequent destabilization near barbed ends. To investigate the mechanisms behind network remodeling, we created a stochastic model with polymerization, depolymerization, branching, capping, uncapping, severing, oligomer diffusion, annealing, and debranching. We find that filament severing, enhanced near barbed ends, can explain the single molecule actin lifetime distribution, if oligomer fragments reanneal to free ends with rate constants comparable to in vitro measurements. The same mechanism leads to actin networks consistent with measured filament, end, and branch concentrations. These networks undergo structural remodeling, leading to longer filaments away from the leading edge, at the +/-35° orientation pattern. Imaging of actin speckle lifetimes at sub-second resolution verifies frequent disassembly of newly-assembled actin. We thus propose a unified mechanism that fits a diverse set of basic lamellipodia phenomenology.

## Editor's evaluation

Although studied for decades, the molecular mechanisms involved in the assembly and remodeling of the lamellipodium still pose a number of questions, among which 1/ how are these networks progressively reorganized from short branched filaments to longer ones while maintaining angular order, 2/ by which mechanisms are actin filaments disassembled in these networks (depolymerization, fragmentation and/or "catastrophic" disassembly), and 3/ what is the importance and contribution of filament annealing? To address these questions, the authors develop one of the most detailed stochastic computational models to date. The model takes into account a large number of chemical reactions, including actin polymerization, depolymerization, filament branching by the Arp2/3 complex, capping, uncapping, severing, oligomer diffusion, annealing, and debranching. Close comparison of in silico and cellular actin networks allows them to evaluate the relative contribution of the different reactions. An important finding of this work is that frequent actin filament severing and annealing are phenomena that cannot be neglected to describe lamellipodial dynamics appropriately and although filament annealing in cells is not a new discovery, it is striking that it is not a negligible and inconsequential phenomenon in the cell, but contributes significantly to the reorganization of actin networks.

**\*For correspondence:** vavylonis@lehigh.edu

**Competing interest:** The authors declare that no competing interests exist.

## Introduction

The force for lamellipodial protrusions is provided by a dendritic network of actin filaments. This dynamic structure is driven by actin filament polymerization, branch generation by the Arp2/3 complex

and regulation of filament elongation by capping protein (*Pollard and Borisy, 2003*; *Watanabe, 2010*; *Blanchoin et al., 2014*). Activated by nucleation promoting factors on the cell membrane, the Arp2/3 complex nucleates filament branches at an angle of approximately 70° from filaments that reach the leading edge. These elongating barbed ends add actin monomers from the cytoplasm to push against the cell membrane and generate force for membrane extension or for the retrograde flow of the whole dendritic actin network toward the cell center. This dendritic lamellipodia network structure, evident in electron micrographs of keratocytes (*Svitkina et al., 1997*) has been quantified by more recent electron tomograms near the leading edge, revealing the number of barbed ends, branches and filaments (*Vinzenz et al., 2012*; *Mueller et al., 2017*). Its characteristic pattern with filaments orientated primarily at ±35° with respect to the protrusion axis (*Maly and Borisy, 2002*; *Schaub et al., 2007*; *Mueller et al., 2017*; *Koseki et al., 2019*) has been interpreted by two-dimensional dendritic network models (*Schaus et al., 2007*; *Maly and Borisy, 2002*; *Weichsel and Schwarz, 2010*; *Atilgan et al., 2005*; *Holz and Vavylonis, 2018*).

In parallel to polymerization and branching at the leading edge, lamellipodia maintain their steady state through continuous disassembly and recycling of actin (*Pollard and Borisy, 2003*; *Watanabe, 2010*; *Blanchoin et al., 2014*). Extensive biochemical and biophysical studies have identified critical aspects of the kinetics and thermodynamics of this turnover process, with cofilin and hydrolysis of ATP bound to actin after polymerization, followed by Pi release, playing a central role. However the precise molecular mechanisms of actin turnover in cells have not been fully resolved (*Danuser and Waterman-Storer, 2006*; *Carlsson, 2010*; *Carlier and Shekhar, 2017*).

Single-Molecule Speckle (SiMS) microscopy of fluorescently labeled actin revealed that actin assembly into the dendritic network is transient and not limited to the leading edge (*Yamashiro et al., 2018*; *Watanabe and Mitchison, 2002*). In these SiMS experiments, actin subunits incorporated into the actin network appear as single molecule speckles while diffuse actin contributes to background fluorescence. In the lamellipodium, speckle disappearances occur within a few seconds after speckle appearances, a time which is relatively short compared to the time required for actin treadmilling through the entire lamellipodium. Since filament treadmilling cannot explain these dynamics, Miyoshi and Watanabe proposed the hypothesis of frequent filament severing near barbed ends, following by annealing of the oligomeric fragment (*Miyoshi and Watanabe, 2013*).

Consistent with the frequent severing near ends and annealing hypothesis, in vitro experiments show dissociation of filament fragments from ends of actin filaments in vitro, in the presence of cofilin and co-factors (*Wioland et al., 2017*; *Kueh et al., 2008*; *Shekhar and Carlier, 2017*; *Andrianantoandro and Pollard, 2006*; *McCullough et al., 2008*). End-to-end annealing of actin filaments is also well-established in vitro (*Sept et al., 1999*; *Andrianantoandro et al., 2001*; *Popp et al., 2007*) as well as in budding yeast (*Okreglak and Drubin, 2010*). Cellular factors such as cofilin and Aip1 may indeed allow filament annealing to the barbed end after severing while also restricting it from resuming elongation (*Okada et al., 2002*; *Wioland et al., 2017*). Additionally, an independent-particle Monte Carlo model based on actin SiMS data used to model FRAP of actin in lamellipodia (*Smith et al., 2013*), as well as actin monomer photoactivation experiments (*Vitriol et al., 2015*), provided better fits with local recycling of slowly diffusing actin back into the network throughout the lamellipodium. A large fraction of slowly diffusing oligomers were indeed observed in fragments of keratocyte lamellipodia (*Raz-Ben Aroush et al., 2017*); however, the model developed by these authors did not consider or require the presence of a local recycling mechanism.

Distributed turnover through severing and annealing may relate to another puzzle of lamellipodia structure revealed in electron micrographs (*Svitkina et al., 1997*): while a dense branched brushwork of filaments is observed near the leading edge (approximately within 1 μm), further away from the leading edge (approximately 3–4 μm away), filaments are longer and appear more linear. The mechanism required for this remodeling has yet to be determined.

To test the hypothesis of frequent severing and annealing in distributed turnover and structural remodeling of the actin network, we created a three-dimensional kinetic model of a steady-state lamellipodium based on the dendritic nucleation model. To develop a network with the observed ±35° filament orientation pattern, we systematically examined the self-organizing filament orientation pattern as a function of the relative network growth speed. Using parameter sets matching lamellipodia of the widely studied keratocyte or XTC cell types, we perform a search over parameters describing uniform severing along the actin filament and enhanced severing near barbed ends. The

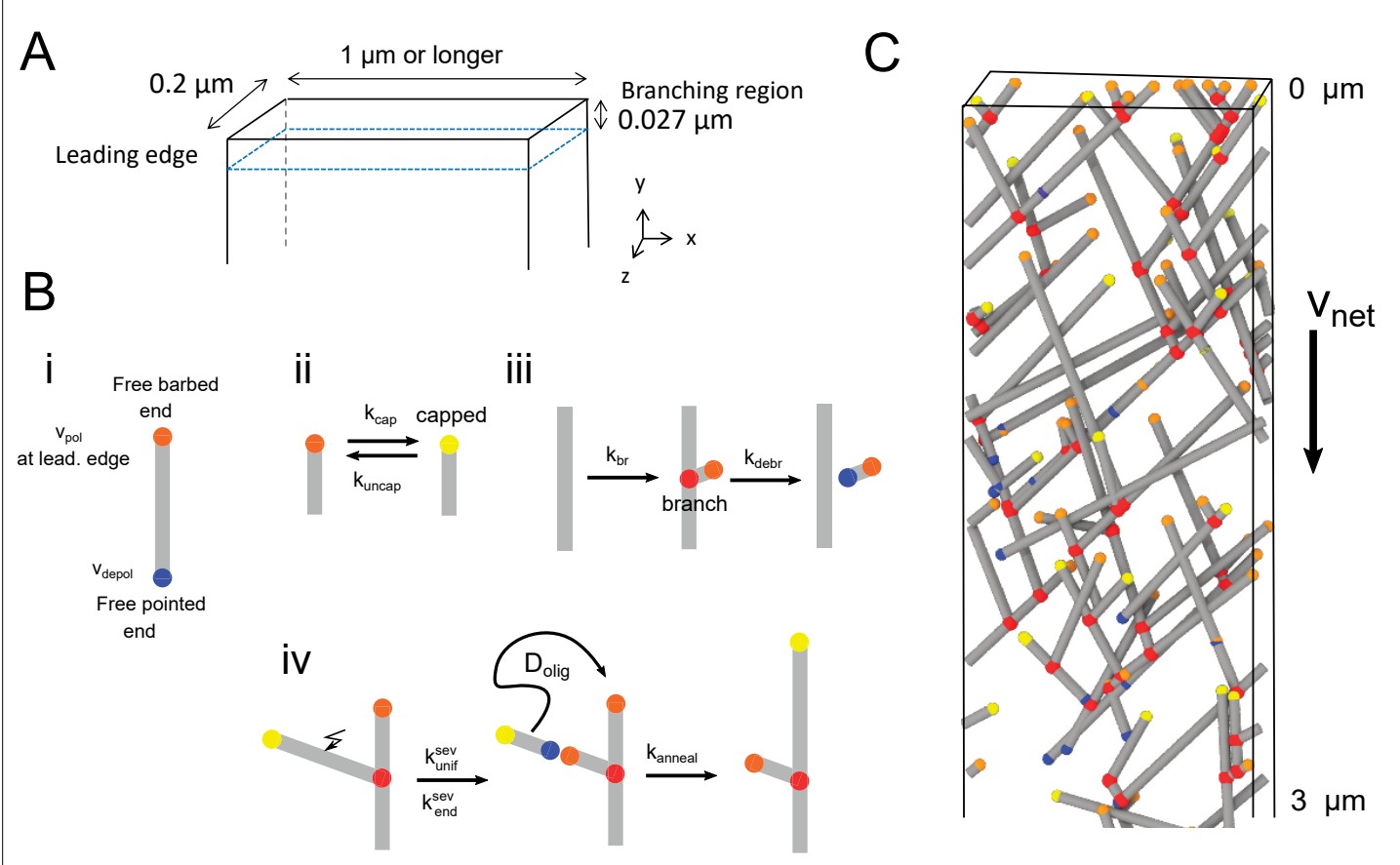

**Figure 1.** Three-dimensional model of the lamellipodial actin network. (**A**) Diagram of the simulation box near the leading edge, which is positioned at $y = 0$ (in the reference frame of the cell) with an open boundary at $y \to \infty$. The thickness of the lamellipodium in the z-direction is 0.2 μm. Periodic conditions are applied along the $x$-direction. Filaments cannot elongate past other boundaries (representing the plasma membrane), where they either stop polymerizing or undergo kinking to elongate along the boundary. (**B**) Cartoons of the processes in the simulation in which filaments are represented as line segments. (**i**) Polymerization at free barbed and depolymerization at free pointed ends. The polymerization rate of free barbed ends away from the leading edge is assumed to occur at a lower rate. (ii) Capping and uncapping of barbed ends. (iii) Branching at 70° occurs along a filament segment within the branching region. (iv) Severing occurs with uniform rate or with a rate enhanced close to barbed ends. If severing results in a fragment of length smaller than $l_{max}^{olig}$, the oligomer fragment is assumed to undergo diffusion with diffusion coefficient $D_{olig}$ (not simulated explicitly). The diffusing oligomer can anneal to a nearby free barbed, or pointed end if the oligomer is uncapped. (**C**) Snapshot of a simulation. Relative speed of the network with respect to leading edge is $v_{net}$. Gray lines: actin filaments; red: Arp2/3 complex; yellow:free barbed ends; orange: capped barbed ends; blue: free pointed ends.

The online version of this article includes the following figure supplement(s) for figure 1:

**Figure supplement 1.** Simulated lamellipodium concentration and depth as a function of uniform severing rate $k_{unif}^{sev}$ and pointed end depolymerization rate $v_{depol}$.

model considers the diffusion and annealing of severed oligomers. We find that a model without annealing cannot reproduce both the filamentous lamellipodial structure and actin SiMS data. With the addition of oligomer diffusion, annealing and enhanced severing near barbed ends, the structure, SiMS data, and overall increase in filament length with distance from the leading edge can be reproduced for optimized parameters. We support this mechanism by performing SiMS of Dylight-labeled actin on XTC cells indicating frequent disassembly of recently polymerized F-actin close to the leading edge. Our study thus supports that frequent severing and annealing is an important mechanism in cellular actin dynamics, motivating further experimental investigations.

**Table 1.** Parameter table for simulations.

| Parameter | Name | Keratocyte Value | XTC Value | Reference/Justification |
|---|---|---|---|---|
| $v_{pol}$ | Polymerization rate at leading edge | 150 sub s$^{-1}$ | 38 sub s$^{-1}$ | Matches observed protrusion rate |
| $v_{depol}$ | Pointed end depolymerization rate | 5 sub s$^{-1}$ | 5 sub s$^{-1}$ | *Wioland et al., 2017*; *Johnston et al., 2015* |
| $k_{cap}$ | Capping rate | 0.6 s$^{-1}$ | 0.2 s$^{-1}$ | Estimated (Materials and methods) |
| $k_{uncap}$ | Uncapping rate | 1.0 s$^{-1}$ | 1.0 s$^{-1}$ | *Miyoshi et al., 2006* |
| $k_{br}$ | Branching rate | 150 s$^{-1}$μm$^{-1}$ | 30 s$^{-1}$μm$^{-1}$ | Estimated (Materials and methods) |
| $k_{debr}$ | Debranching rate | 0.1 s$^{-1}$ | 0* | Narrower distribution of Arp2/3 complex compared to F-actin (*Lai et al., 2008*, *Miyoshi et al., 2006*, *Ryan et al., 2012*) |
| $v_{net}$ | Network velocity with respect to leading edge | 0.2μm s$^{-1}$ | 0.05μm s$^{-1}$ | |
| $k_{unif}^{sev}$ | Uniform severing rate | Varied | Varied | |
| $k_{end}^{sev}$ | Severing rate near barbed end | Varied | Varied | |
| $k_{anneal}$ | Annealing rate constant | 60μM$^{-1}$s$^{-1}$ | 60μM$^{-1}$s$^{-1}$ | Close to *Popp et al., 2007* |
| $l_{max}^{olig}$ | Maximum oligomer size | Varied | Varied | |
| $D_{olig}$ | Oligomer diffusion coefficient | 0.25μm$^2$ s$^{-1}$ | 0.25μm$^2$ s$^{-1}$ | Estimated |

*Since severing and depolymerization contributed to debranching in XTC cells, we did not include a separate debranching rate constant.

## Results

### Stochastic simulation of dendritic network

The model shown in *Figure 1* includes barbed end polymerization, pointed end depolymerization, capping, uncapping, branching near the leading edge, debranching, severing and annealing, without explicitly considering ATP hydrolysis or phosphate release (see Materials and methods and *Table 1*). We impose a constant network velocity $v_{net}$ with respect to the leading edge. We selected parameters corresponding to two frequently-studied cell systems, one for fast moving keratocyte cells and one for XTC or fibroblast cells. For keratocytes, $v_{net}$ corresponds to the rate of cell protrusion, since the actin network is almost stationary with respect to the substrate in experiments (*Keren et al., 2008*; *Mueller et al., 2017*; *Yamashiro et al., 2018*; *Schaub et al., 2007*). Lamellipodia of XTC cells are frequently studied in cells that do not crawl on the substrate, so $v_{net}$ provides the magnitude of the retrograde flow speed (*Watanabe and Mitchison, 2002*; *Ryan et al., 2012*). We considered uniform severing with rate constant $k_{sev}^{unif}$ per filament length and enhanced end severing with a rate per filament $k_{sev}^{end}$ near the barbed end. If an oligomer is created from one of these severing events, we assume it can diffuse and anneal to a nearby filament end with rate constant $k_{anneal}$.

### Planar branching along lamellipodial plane sharpens the filament orientation pattern

Prior models of dendritic networks demonstrated how the ±35° orientation with respect to the direction of protrusion depends on the relationship between filament elongation velocity $v_{pol}$ and relative extension rate $v_{net}$. As shown in the results of the 2D model by *Weichsel and Schwarz, 2010* in *Figure 2A*, for low $v_{net}/v_{pol}$, the dominant orientation pattern has filaments branching at $-70°/0/70°$. Filaments oriented at angles larger than $\theta_c$, for which $\cos(\theta_c) = v_{net}/v_{pol}$, lose contact with the membrane since they are not polymerizing quick enough to catch up. When this critical angle becomes smaller than 70°, the favored pattern is filaments with orientations centered at ±35°: the filament population around 35° can generate daughter branches at -35° and vice versa; thus the population sustains itself even as individual filaments get capped. The $-70°/0/70°$ was found to

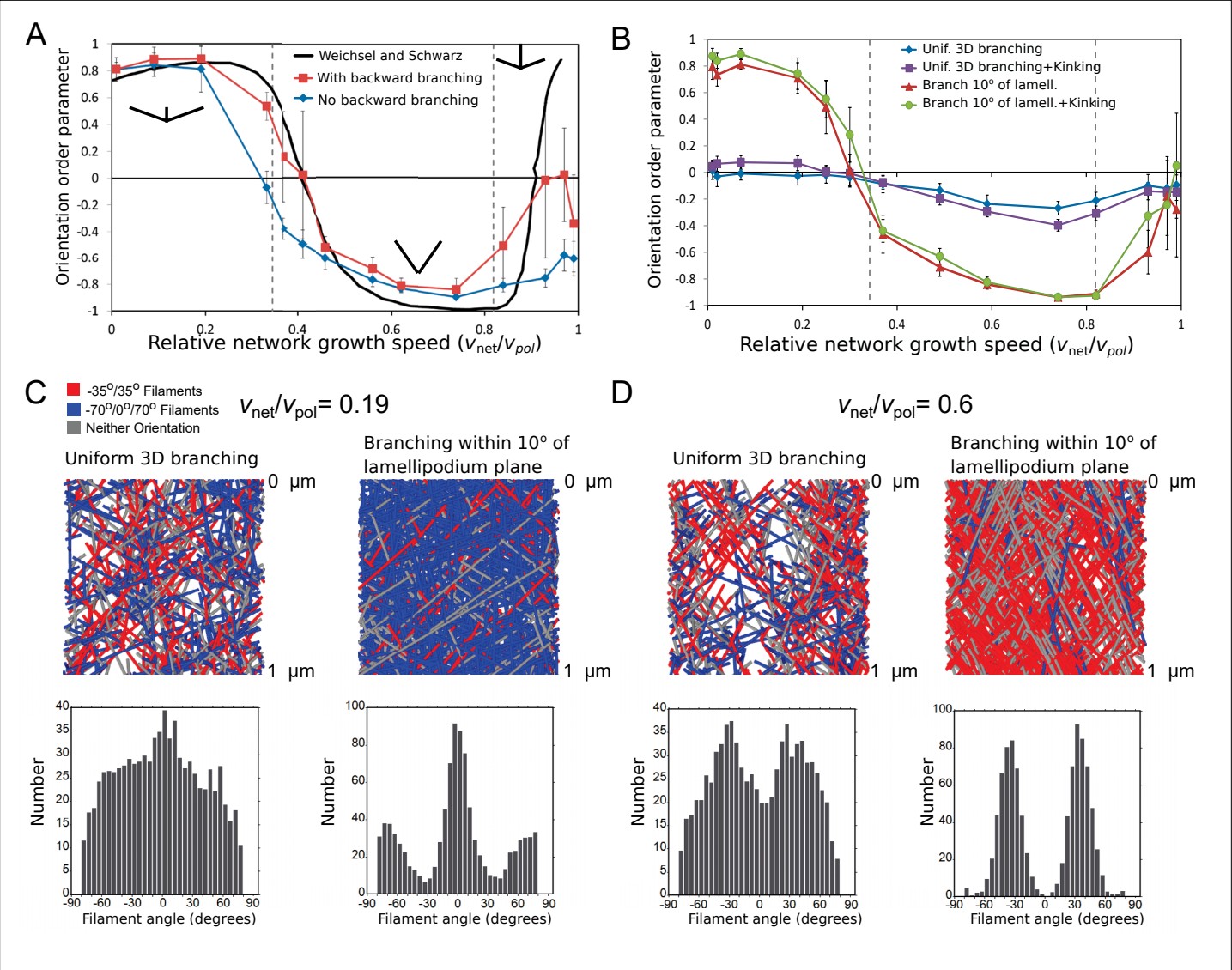

**Figure 2.** Steady-state filament orientation patterns in 2D and 3D and dependence on planar branching restriction. (**A**) Orientation order parameter as a function of relative network growth speed, $v_{net}/v_{pol}$ in a simulation where branching always occurs along the lamellipodium plane (within 1°). Numerical data with or without backward branching are compared to numerical results in 2D without backward branching from Figure S1A in **Weichsel and Schwarz, 2010**. For the plots, $v_{net} = 0.05~\mu m/s$ was constant and $v_{pol}$ was varied. An orientation order parameter equal to 1 indicates a network with all filaments in the $-70°/0°/70°$ orientation, while –1 indicates the ±35° orientation. Vertical dashed lines indicate the critical $v_{net}/v_{pol}$ for filaments at 70° and 35° along the lamellipodium plane. (**B**) Same as panel A but for a 3D simulation in which 70° branching occurs at random orientation or uniformly within 10° of the lamellipodium $xy$ plane, without backward branching. The filament orientation is calculated from the filament projection along the $xy$ lamellipodium plane. Uniform branching leads to a less ordered network, even when kinking of filaments hitting a boundary is implemented. (**C**) Top: 3D simulation snapshots, colored by orientation pattern, for $v_{net}/v_{pol} = 0.19$ (no kinking). Bottom: orientation distribution of filaments with a portion located within 1 μm of the leading edge, average of 5 simulations reaching steady state. Restricting branching along the lamellipodium plane sharpens the $-70°/0°/70°$ orientation pattern. (**D**) Same as panel C, for $v_{net}/v_{pol} = 0.6$ (no kinking), showing that restricting branching along the lamellipodium plane sharpens the ±35° orientation pattern. Parameters are as listed in **Table 1** (keratocyte parameter set) with $k_{unif}^{sev} = 10^{-5}$ sub/s, but with no oligomer annealing or enhanced end severing: $k_{anneal} = k_{end}^{sev} = l_{max}^{olig} = 0$.

The online version of this article includes the following figure supplement(s) for figure 2:

**Figure supplement 1.** Addition of kinking does not effect the orientation pattern.

resume when $v_{net}/v_{pol}$ exceeds the critical angle for 35°; in this situation, only the 0° filaments polymerize quick enough to remain in the branching region. The above behavior can be quantified by the orientation order parameter, where a value 1 (or –1) indicates all filaments are in a –70°/0°/70° (or ±35°) pattern.

By contrast to the above 2D results, dendritic network models formulated in 3D have provided apparently contradictory results (*Holz and Vavylonis, 2018*). *Atilgan et al., 2005* reported that obtaining the ±35° pattern requires restricting branching to occur primarily along the lamellipodium plane, which they attributed to structural constraints of the branching machinery at the leading edge. *Schreiber et al., 2010* and *Hu and Papoian, 2010* however did observe the ±35° pattern in 3D simulations, but the role of $v_{net}/v_{pol}$ in determining the pattern was not examined.

As we are interested in structural aspects of lamellipodia, we performed systematic simulations in both 2D and 3D to examine the filament orientation pattern as function of $v_{net}/v_{pol}$. In the simulations of *Figure 2* we consider the region close to the leading edge where severing, annealing and debranching does not influence the resulting structure, and varied the polymerization rate while keeping $v_{net}$ constant. We also kept the capping and branching rates $k_{cap}$ and $k_{br}$ fixed as the orientation pattern is robust with respect to their values (*Weichsel and Schwarz, 2010*).

We reproduce the results of the *Weichsel and Schwarz, 2010* simulations as a function of relative network growth speed, by imposing a tight planar branch restriction along the lamellipodial $xy$ plane (*Figure 2A*). Note that the transition among different orientation patterns is not abrupt at the critical angles, as a result of allowing fluctuations in branching angle and a finite size of the branching region. This is the reason for the largest difference occurring at high $v_{net}/v_{pol}$: because only filaments at small angles can keep up with the leading edge under these conditions, our simulation evolves to a narrow comet-like branching structure where all branching is concentrated; this allows branching among ±35° and other orientations (e.g. $-80°/-10°/60°$) to be maintained through double branching before filaments exit the branching region.

There is little difference in the orientation order parameter between simulations that allow backward branching (angles greater than 80°, chosen to include angles that would be oriented away from the leading edge considering the variation of branching angle for both orientation patterns) and branching limited toward the leading edge (*Figure 2A*). Filaments that branch backward exit the branching region quicker than filaments that branched forward, decreasing the likelihood of branches off of backward filaments. Even though backward-facing filaments contribute to angles larger than 80°, they do not influence the order parameter that does not measure them. Since backward branching does not affect the filament orientation pattern, and since backward-facing filaments are not seen in electron microscopy images (*Vinzenz et al., 2012*; *Mueller et al., 2017*), backward branching was not allowed in the rest of our simulations.

Next, we allowed branching to occur uniformly in 3D (i.e. with equal probability along any azimuthal angle with respect to the axis of the parent filament) and studied the orientation order parameter as function of $v_{net}/v_{pol}$ (*Figure 2B*). The order parameter was measured using the angles of filaments projected along the lamellipodial $xy$ plane. Uniform branching led to low and weakly varying order parameter, unlike the sharp orientation pattern with distinct transitions observed in 2D. By comparison, a quasi-2D simulation with branching allowed to occur within 10° off the lamellipodial plane (close to the maximum off-plane angles observed in electron microscopy [*Vinzenz et al., 2012*] [A. Narita, personal communication, March 2018]) restored the behavior observed in 2D.

In the simulations described so far, filament elongation was assumed to stop when the polymerizing barbed end reached the top or bottom $z$ plane. To investigate the influence of filament bending along the membrane, we also performed simulations with filament 'kinking', in which filaments were allowed to continue their elongation parallel to the plane representing the top or bottom cell membrane. Allowing kinking lead to a high concentration of filaments along the top and bottom plane of the lamellipodium but did not however significantly influence the orientation order parameter (*Figure 2B*).

To further visualize the network structure in simulations, *Figure 2C and D* (and *Figure 2—figure supplement 1* for the case with kinking) show snapshots and filament orientation patterns at low and intermediate values of relative network growth speeds. The simulations with quasi-2D branching show clear –70°/0°/70° and ±35° respective orientation patterns. Interestingly, even though no prominent features are observed in simulations with uniform 3D branching at low relative network speeds

(*Figure 2C*), intermediate relative network speeds do show features at ±35° (*Figure 2D*). The latter histogram is not very different from experimentally-measured distributions (*Maly and Borisy, 2002*; *Vinzenz et al., 2012*; *Koseki et al., 2019*).

We thus conclude that the ±35° pattern does occur within a broad range of relative network growth speeds with uniform 3D branching, however the peaks at ±35° are not very pronounced. Indeed, the parameters used by *Atilgan et al., 2005* corresponded to $v_{net} = 0.26 - 0.46$, a parameter set that mostly lies outside the ±35° region; this is likely the reason why the ±35° was not observed in this study. Our results also suggest why *Schreiber et al., 2010* who used $v_{net} \approx 0.37$ and *Hu and Papoian, 2010* who had $v_{net} = 0.45 - 0.51$, did observe a ±35° with uniform 3D branching.

Considering the simulation results as well as experimental evidence in electron tomograms for filaments oriented primarily along the lamellipodial plane (*Vinzenz et al., 2012*), for the rest of the simulations we proceed with the quasi-2D case where filament branching occurs within 10° of the lamellipodial plane and relative network growth speeds result in a ±35° orientation pattern.

## Estimated parameters

For both cases of keratocytes and XTC cells, which correspond to different values for $v_{net}$, we estimated the rates of polymerizing barbed end elongation $v_{pol}$, branching $k_{br}$, and capping $k_{cap}$, that are needed for a dendritic network with the anticipated concentration, branch length, and ±35° filament orientation pattern (*Table 1*). We also assume that barbed ends can uncap with rates comparable to those in SiMS lifetime measurements of capping protein (*Miyoshi et al., 2006*). The availability of such uncapped barbed ends for annealing is an important assumption of this work. We further use debranching rates by considering the lifetime of Arp2/3 complex components in SiMS (*Miyoshi et al., 2006*) and measurements of Arp2/3 complex profiles in lamellipodia (*Iwasa and Mullins, 2007*; *Lai et al., 2008*; *Ryan et al., 2012*).

We also assumed that uncapped barbed ends away from the leading edge do not elongate or shrink and that free pointed ends depolymerize with a rate $v_{depol} = 5/s$. The results we present below are robust with respect to small changes of these parameters, as long as the overall filament disassembly rate away from the leading edge is not reaching values comparable to $v_{pol}$. Maintaining a wide lamellipodium in the latter case would require a global treadmilling mechanism, which would contradict the evidence for distributed turnover. The assumption of slow barbed end dynamics away from the leading edge is consistent with the slow intensity increase in the back of lamellipodium after FRAP of actin (*Smith et al., 2013*) or after photoactivation of actin at the cell middle (*Lai et al., 2008*; *Vitriol et al., 2015*), as well as evidence that cofilin and twinfilin promote both barbed and pointed end depolymerization (*Wioland et al., 2017*; *Johnston et al., 2015*; *Hakala et al., 2021*; *Shekhar et al., 2021*).

A steady state with a finite lamellipodium depth is reached in the simulation whenever the net rate of depolymerization balances the net rate of polymerization at the leading edge. For example, in the case without enhanced end severing, annealing, or oligomer dissociation, the depth of the lamellipodium is determined by the rates of uniform severing, $k_{unif}^{sev}$, and $v_{depol}$ (*Figure 1—figure supplement 1*): in this case, the fast growth of barbed ends at the leading edge is balanced by the slower depolymerization of a larger number of pointed ends created by severing.

Given the parameters in *Table 1*, this leaves three main unknown parameters related to oligomer dissociation: $k_{unif}^{sev}$, the rate of uniform severing along each filament; $l_{max}^{olig}$, the longest length of a diffusing oligomer; $k_{end}^{sev}$, the enhanced severing rate near the barbed end (within $l_{max}^{olig}$ of the end). We treated these three as fitting parameters and considered separately the cases in the presence or absence of annealing. In the absence of annealing, dissociating oligomers do not reincorporate into the network and are thus discarded from the simulation (corresponding to eventual disassembly into monomers, a process that we did not simulate).

## Model without annealing cannot reproduce both actin SiMS data and lamellipodial structure

We conducted a parameter search over the maximum oligomer size $l_{max}^{olig}$ as well as end and uniform severing rates $k_{end}^{sev}$, $k_{unif}^{sev}$ for keratocyte- and XTC-like parameters without annealing (*Figure 3—figure supplement 1*). We classified each parameter set in terms of how well it described the F-actin structure and concentration (including barbed end and branch concentration profiles), actin SiMS data

(speckle lifetime distribution, appearance and disappearance profile), and if there was an increase in length between the filaments located in 0–1 µm and 3–4 µm region (see Materials and methods). Within a range of uniform and end severing rates, and short maximum oligomer lengths, the network structure and concentration of our simulation was close to that expected for keratocyte and XTC lamellipodium (*Figure 3—figure supplement 1*). However, as expected, we did not find any parameter sets that resulted in a length increase since a mechanism for an increase in length and remodeling is not included. We also did not find any parameter set with a sharp peak at short actin speckle lifetimes, though in some cases the actin speckle lifetime distribution has a peak at relatively short times (*Figure 3—figure supplement 1*).

To further demonstrate that the model without annealing cannot fit the experimental data, *Figure 3* and *Figure 3—figure supplement 2* contain examples of results for keratocyte and XTC parameters. In this and subsequent figures we color parameter sets black without end severing and red that include enhanced end severing; these curves correspond to the parameters of the scan that are highlighted with a thick frame of same color in *Figure 3—figure supplement 1*. For the case with moderate uniform severing and no end severing, both the XTC and keratocyte uniform severing cases have a concentration profile that is comparable to lamellipodium of their respective parameter set (*Figure 3*) as well as barbed end and branch distributions (*Figure 3—figure supplement 2*). However, the actin speckle lifetimes do not peak at short lifetimes. With the addition of enhanced end severing, as well as increase of uniform severing, a peak at short lifetimes is observed that is closer to the experimental SiMS curves; however, the lamellipodium becomes too narrow, there is a shortage of long speckle lifetimes compared to experiment, and the location of speckle appearances is restricted close the leading edge (*Figure 3* and *Figure 3—figure supplement 2* A,F). In conclusion, the model without annealing cannot reproduce the distributed turnover, structure of actin network and increase in filament length.

## Model with oligomer annealing can reproduce lamellipodium structure, actin speckle dynamics and increase in filament length away from leading edge

Next, we examined if inclusion of oligomer annealing might be able to provide an adequate fit to the structure, speckle, and length increase criteria. We performed another parameter search over the maximum oligomer length, end and uniform severing rates ($l_{max}^{olig}$, $k_{end}^{sev}$, $k_{unif}^{sev}$) for keratocyte and XTC parameter sets. For these simulations, we used an annealing rate constant measured in crowded surfaces in vitro (*Popp et al., 2007*). Similar to the parameter scan without annealing, a match to F-actin structure and concentration was obtained when uniform and end severing rates were within a certain range (*Figure 4—figure supplement 1* and *Figure 5—figure supplement 1*). The addition of annealing improved the agreement between the speckle lifetimes, appearance and disappearance location profiles compared to SiMS data, resulting in parameter sets that agree with both structure and speckle dynamics (*Figure 4—figure supplement 1* and *Figure 5—figure supplement 1*). We also see that with high enough severing and maximum oligomer lengths, the filament length increases in a region away from the leading edge (*Figure 4—figure supplement 1* and *Figure 5—figure supplement 1*). Parameter sets where all three fitting criteria are satisfied exist for both keratocyte and XTC cases, and all these triple matches have a finite enhanced end severing rate.

Detailed results from our parameter scan for keratocytes are shown in *Figure 4*. This figure shows a case without enhanced end severing (black curves, double match in structure and speckles for keratocyte parameters) and with enhanced end severing (red, triple match in structure, speckles and filament length increase). Both parameter sets can reproduce the actin speckle dynamics as seen in the speckle lifetime, appearance and disappearance location distributions as well as the the structure as seen in the F-actin, barbed end, and branch concentration profiles (*Figure 4A–F*). However, only the case with enhanced end severing results in a simultaneous increase in length away from the leading edge (*Figure 4G*). The increase brings the filament length close to the estimated average of 800 nm in keratocytes (*Schaub et al., 2007*). The concentration profile of Arp2/3 complex branches (*Figure 4F*), snapshots (*Figure 4H*), and *Video 1* of the optimized simulations with enhanced end severing clearly show more branches and shorter filaments in the region near the leading edge than away from the leading edge, similar to the electron micrographs of *Svitkina et al., 1997*. The ±35° filament orientation pattern is preserved throughout the lamellipodium in

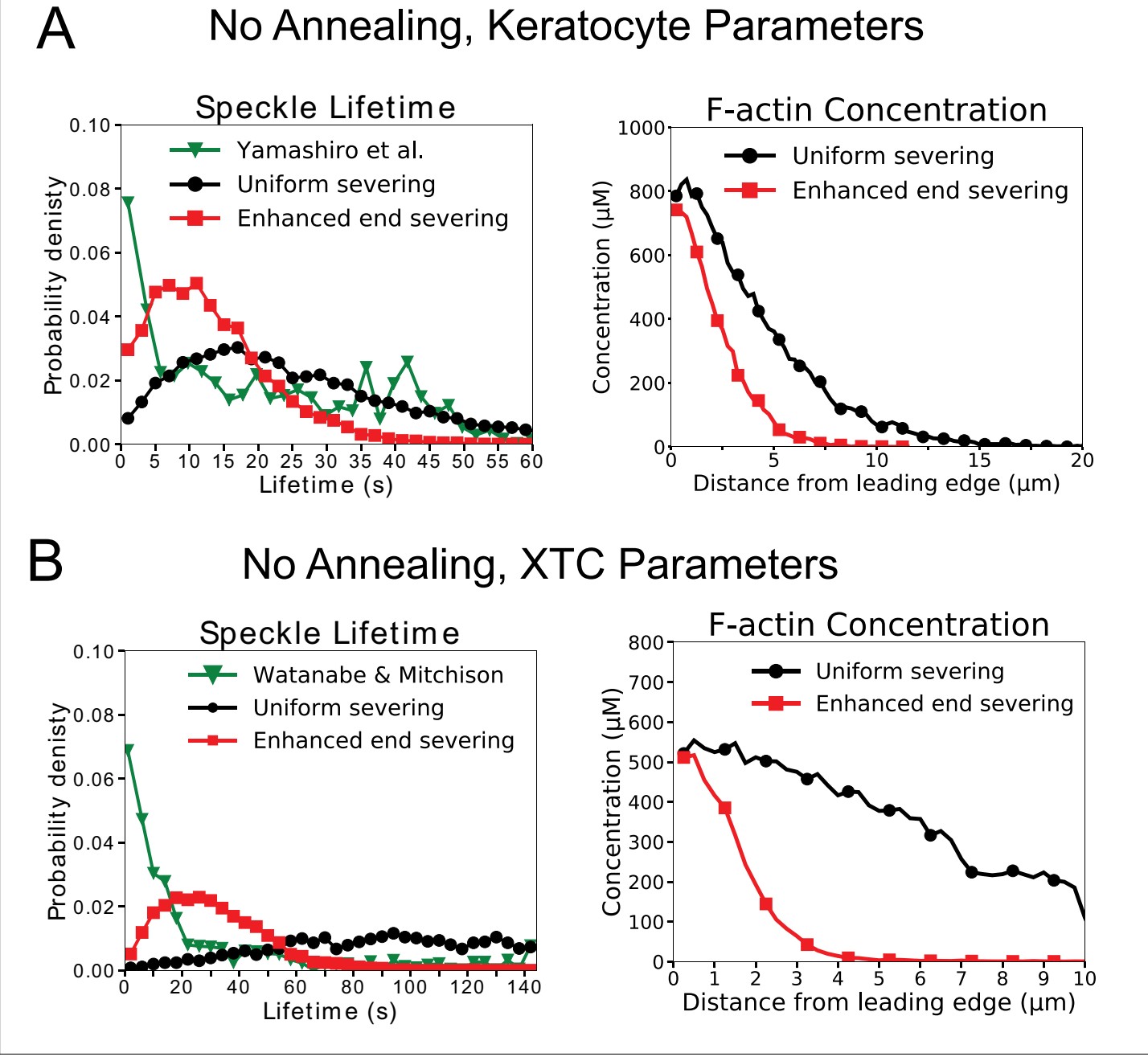

**Figure 3.** The model without annealing cannot reproduce both the actin speckle lifetime and F-actin concentration profiles. (**A**) Comparison of optimized parameters of model with uniform severing or model with enhanced barbed end severing to experiments in keratocytes. Left: Probability density of simulated actin speckle lifetimes and comparison to SiMS measurements in *Yamashiro et al., 2014*. Distributions were normalized between 2 and 60 s, to exclude short lifetimes beyond experimental resolution. Right: F-actin concentration profile for keratocyte parameters. Keratocyte parameters as in *Table 1* with $k_{unif}^{sev} = 10^{-4}$ /sub/s; $l_{max}^{olig} = 40$ sub (black) and $k_{unif}^{sev} = 5 \cdot 10^{-4}$ /sub/s; $k_{end}^{sev} = 5 \cdot 10^{-4}$ /sub/s; $l_{max}^{olig} = 80$ sub (red). Increasing the severing rate near the barbed end to better match the short speckle lifetime experimental peak leads to short lamellipodium. (**B**) Comparison of optimized parameters of model with uniform severing or model with enhanced barbed end severing to experiments XTC cells by *Watanabe and Mitchison, 2002* (as they were corrected for photobleaching). Same as panel A, with probability density of speckle lifetimes normalized between 4 and 144 s. XTC parameters as in *Table 1* with $k_{anneal} = 0$, and $k_{unif}^{sev} = 10^{-5}$ sub/s; $l_{max}^{olig} = 40$ sub (black) or $k_{unif}^{sev} = 5 \cdot 10^{-4}$ /sub/s; $k_{end}^{sev} = 10^{-4}$ /sub/s; $l_{max}^{olig} = 80$ sub (red). Concentration profiles are the average of 5 simulations in steady state. Speckle lifetimes measured for speckles within 12 μm of the leading edge over a 20 s interval in steady state for 5 simulations.

The online version of this article includes the following figure supplement(s) for figure 3:

*Figure 3 continued on next page*

**Figure supplement 1.** Parameter scan of uniform severing and enhanced end severing rates ($k_{unif}^{sev}$, $k_{end}^{sev}$) and maximum oligomer length ($l_{max}^{olig}$) without annealing.

**Figure supplement 2.** Quantification of model without annealing that did not reproduce the actin speckle lifetime and F-actin concentration profiles (continuation of *Figure 3*).

the presence of severing and annealing. Additional quantification in *Figure 4—figure supplement 2* shows the steady state spatial distribution of capped and uncapped barbed ends, the approximate uniform distribution of oligomer sizes between 0 and $l_{max}^{olig}$, and the profile of F-actin according the mechanism of assembly (polymerization as monomer at the leading edge versus annealing). Most of the F-actin at the back of the lamellipodium has undergone severing and annealing (*Figure 4—figure supplement 2D.F*), similar to an earlier particle model for distributed turnover (*Smith et al., 2013*).

For XTC cells, enhanced end severing is needed to match the actin speckle lifetime distribution, which further contributes to filament length increase. Results from our parameter scan for XTC cells in *Figure 5* show a case without enhanced end severing (black curves, match in structure only) and a case with enhanced end severing (red, triple match in structure, speckles and filament length increase). The case with enhanced end severing provides a good overall fit to SiMS and structure (*Figure 5A–G*), demonstrating increase in length away from the leading edge (*Figure 5G and H* and *Video 2*). *Figure 5B and F* show that the model reproduces the narrower distribution of Arp2/3 complex as compared to F-actin in XTC cells (*Ryan et al., 2012*). Additional quantification in *Figure 5—figure supplement 2* shows profiles of capped and uncapped end, oligomer size distribution, and origin of F-actin.

When performing the F-actin concentration and structure match for the XTC parameter sets, we used numbers in between those of electron tomograms in fibroblasts by *Vinzenz et al., 2012* and the estimated F-actin concentration of 1000 μM for XTC cells (*Watanabe, 2010*). The branching rate in the simulations of *Figure 5* corresponds to 0.05 μM s$^{-1}$, which is about half of the Arp2/3 complex nucleation rate of 0.11 μM s$^{-1}$ estimated by SiMS (*Watanabe, 2010*; *Miyoshi et al., 2006*). We checked that simulations with doubled the branching rate still provide a good fit to actin SiMS data as well as a length increase away from the leading edge, with F-actin concentration at the leading edge that was around 1,100 μM (*Figure 5—figure supplement 3*). We also tested that excluding end severing of polymerizing ends in *Figures 4 and 5* did not modify our results for the optimized parameter sets (*Figure 5—figure supplement 4*).

The optimized parameter sets with speckle dynamics similar to SiMS experiments of *Figures 4 and 5* also matched another observation from SiMS (*Watanabe and Mitchison, 2002*): the distribution of actin speckle lifetimes was weakly dependent on location of appearance with respect to the leading edge (*Figure 4—figure supplement 3*, *Figure 5—figure supplement 5*).

The simulations of *Figures 4 and 5* implement a mechanism of local oligomer rebinding, which is needed to match actin FRAP and photoactivation data (*Smith and Liu, 2013*; *Vitriol et al., 2015*): as shown in *Figure 4—figure supplement 3B* and *Figure 5—figure supplement 5B* the distance travelled by oligomers before annealing is in the sub-μm range. As a further check of consistency of our simulations with SiMS results, simulated actin SiMS for parameters with enhanced end severing of *Figure 4* (keratocytes) and *Figure 5* (XTC) do resemble experimental images from *Yamashiro et al., 2014* and *Watanabe and Mitchison, 2002* (*Videos 3 and 4*). In *Videos 3 and 4*, when an appearing and disappearing speckle are near one another, this is typically a reannealing event occurring quickly over short distances. We confirmed that events that might be limited by spatial and temporal resolution or interpreted as blinking in SiMS experiments correspond to a very small fraction of speckle appearances in the simulation.

Finally, we note that even though the results of this section were obtained for a specific value of annealing rate constant and oligomer diffusion coefficient, they remain valid as long as the annealing of oligomers occurs over a sufficiently short distance. We find that this is the case even for annealing rate constants that are lower by nearly two orders of magnitude compared to the values of *Table 1* (*Figure 4—figure supplement 3C*, *Figure 5—figure supplement 5C*).

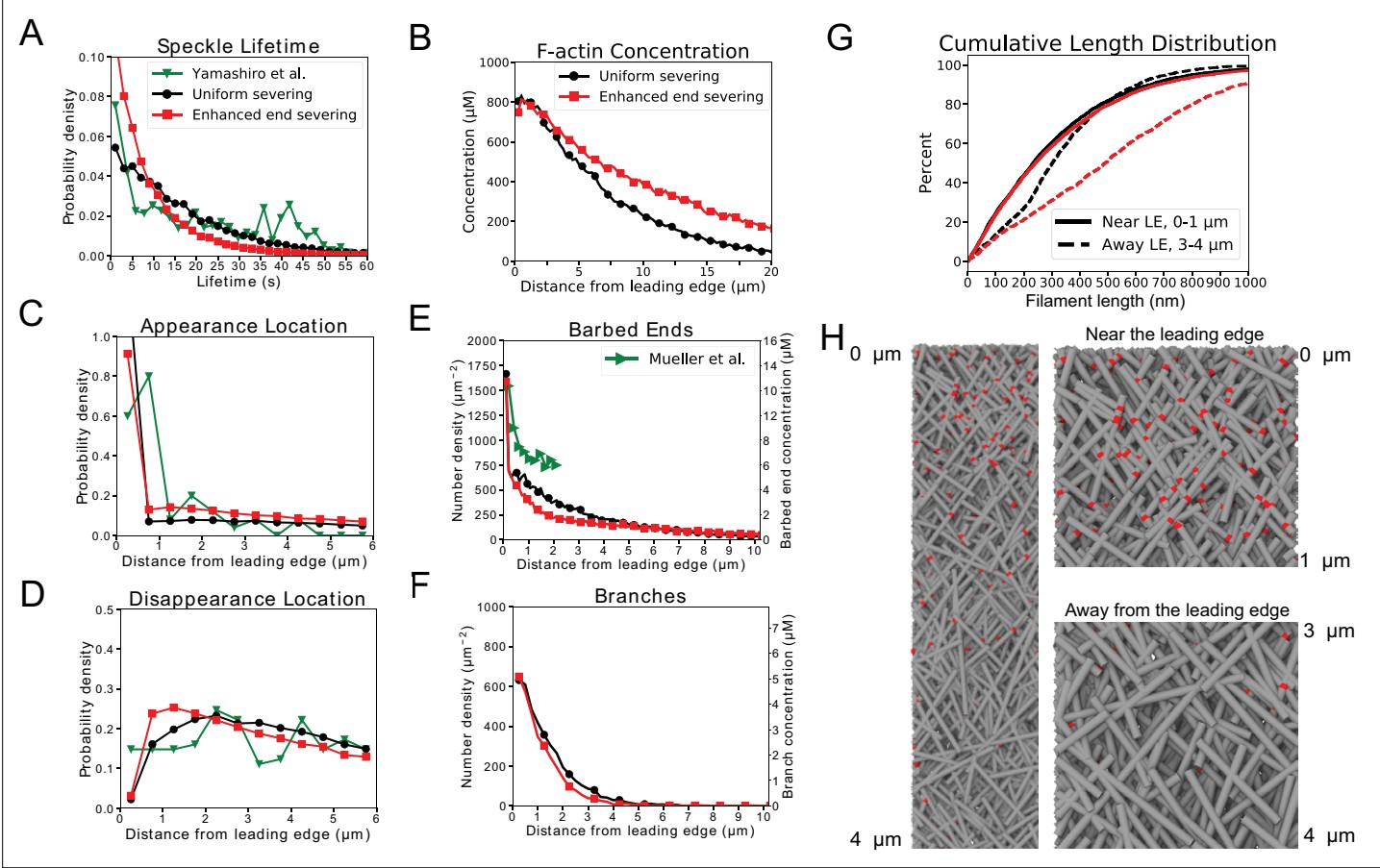

**Figure 4.** Model results for optimized parameters of model for keratocytes with uniform severing and annealing (black) or model with annealing and severing, enhanced near barbed end (red). The latter case provides good agreement with SiMS data, actin network structure, and filament length increase away from the leading edge. (**A**) Probability density of simulated actin speckle lifetimes and comparison to SiMS measurements in *Yamashiro et al., 2014*. Distributions were normalized between 2 and 60 s, to exclude short lifetimes beyond experimental resolution. (**B**) F-actin concentration profile. The oligomeric actin concentration (within 0–10 μm) was less than 0.1% of F-actin in that region. (**C**) Simulated actin speckle appearance location and comparison to *Yamashiro et al., 2014*. Distributions were normalized within the indicated range, considering speckles with lifetimes longer than 2 s. (**D**) Same as C, for disappearance location. (**E**) Distribution of barbed ends and comparison to measurements in *Mueller et al., 2017*. The experimental data are plotted according to the left y-axis. The concentrations on the right $y$-axis use the model's assumed lamellipodial thickness. (**F**) Concentration of simulated Arp2/3 complex branches. (**G**) Cumulative filament length distributions near $(0 - 1\mu m$, solid) and away from the leading edge $(3 - 4\mu m$, dashed). (**H**) Snapshot of simulation with enhanced end severing (left). Zoomed in views close and away from the leading edge (right). Lamellipodium width is 1 μm. Gray lines: actin filaments; red: Arp2/3 complex. Parameters are listed in *Table 1* (keratocyte parameters). The simulation with uniform severing used $k_{unif}^{sev} = 5 \cdot 10^{-4}$ /sub/s; $l_{max}^{olig} = 80$ sub and with enhanced end severing $k_{unif}^{sev} = 5 \cdot 10^{-4}$ /sub/s; $k_{end}^{sev} = 1 \cdot 10^{-3}$ sub/s; $l_{max}^{olig} = 150$ sub. Data averaged over 5 independent simulations. Speckle data measured for speckles within 12 μm of the leading edge over a 20 s interval in steady state for 5 simulations.

The online version of this article includes the following figure supplement(s) for figure 4:

**Figure supplement 1.** Parameter scan of uniform and enhanced end severing rates ($k_{unif}^{sev}$, $k_{end}^{sev}$) and maximum oligomer length ($l_{max}^{olig}$), with oligomer annealing but no debranching $k_{debr} = 0$.

**Figure supplement 2.** Additional quantification of model results for keratocytes with annealing and uniform severing (top row), or annealing and severing enhanced near barbed ends (bottom row).

**Figure supplement 3.** Quantification of actin turnover in simulations with severing and annealing of keratocyte parameters from 5 simulations in steady-state over 20 s each.

**Figure supplement 4.** Results of alternate model for keratocytes with frequent barbed end depolymerization followed by rapid repolymerization (*Video 5*).

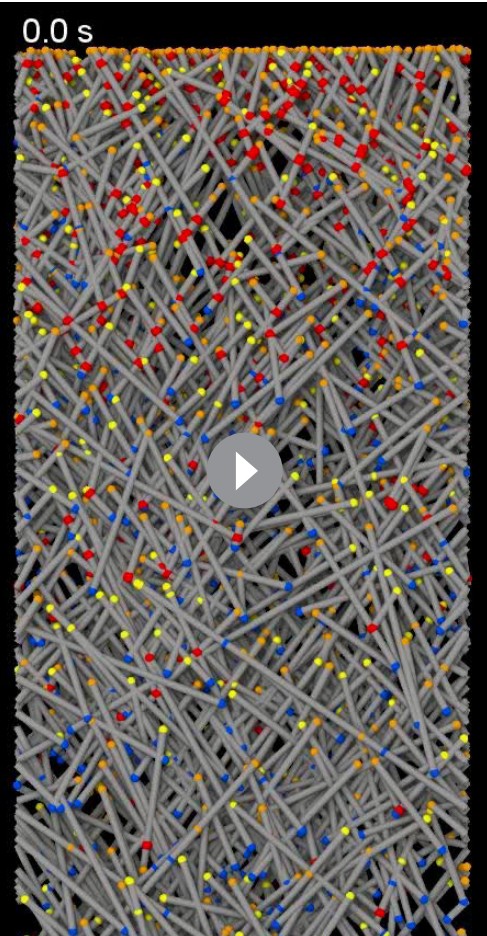

**Video 1.** Simulated keratocyte lamellipodium. Video of simulated keratocyte lamellipodium in the rest frame of the cell (0–4 µm with the leading edge located at the top) for the enhanced end severing parameter set (Figure 4). Oligomer fragments not shown. Fragments of filaments that appear correspond to annealing events and fragments that disappear to the creation of oligomers. End severing events are shown by disappearance of fragments near barbed ends. Uniform severing events can be identified by the appearance of pairs of pointed and barbed ends. Some filaments overlap one another as we do not have excluded volume interactions. Few filaments near the leading edge can be seen polymerizing after uncapping. These barbed ends were annealed to by a polymerizing oligomer created by uniform severing. The barbed end state of the oligomer was transferred to the filament resulting in a polymerizing filament away from the leading edge. Gray lines: actin filaments; red: Arp2/3 complex; yellow:free barbed ends; orange: capped barbed ends; blue: free pointed ends. Each frame is 0.1 s. Leading edge is 2 µm wide.

https://elifesciences.org/articles/69031/figures#video1

## Alternative mechanism with frequent barbed end destabilization

An alternative mechanism to explain the short actin SiMS lifetimes (different to enhanced severing and annealing) is barbed end catastrophic disassembly by factors such as twinfillin (*Wioland et al., 2017*; *Johnston et al., 2015*; *Hakala et al., 2021*; *Shekhar et al., 2021*) or cooperative strand separation in the presence of cofilin, coronin and Aip1, depending on the cofactor concentrations (*Kueh et al., 2008*; *Jansen et al., 2015*; *Tang et al., 2020*).

To test such a mechanism, we implemented a model with stochastic transitions to rapid barbed end depolymerization (see Materials and methods, *Video 5*). To maintain the F-actin loss by such a disassembly process, a process of rapid regrowth must also be included. Using the insight gained from our parameter searches with severing and annealing, we can show that such a model can be tuned to come close to matching our three main experimental test criteria: agreement with SiMS data, an increase in filament length with distance from the leading edge, and broad F-actin concentration profile (*Figure 4—figure supplement 4*).

While we cannot fully exclude such a dynamic-instability-like mechanism, we note that: (1) it would require additional controls or homeostatic mechanisms to balance disassembly and reassembly away from the leading edge (while maintenance of F-actin mass is ensured by a severing and annealing mechanism), (2) ATP hydrolysis associated with ATP-actin monomer polymerization would be energetically more costly, and (3) reassembly of rapidly diffusing monomers away from the leading edge to recover bursting would also be less consistent with FRAP or photoactivation experiments (*Smith et al., 2013*; *Vitriol et al., 2015*).

## Short actin speckle lifetimes provide evidence for rapid disassembly near barbed ends

The actin speckle lifetime distribution of the models with severing and annealing of *Figures 4–5* characteristically peaks at $t = 0$. This is a general feature of a barbed-end disassembly mechanism where newly polymerized monomers are the ones that disassemble at higher rates, being closer to the barbed end. The experimental SiMS lifetimes in *Figures 3–5* used a temporal resolution of 1 s (*Watanabe and Mitchison, 2002*; *Yamashiro et al., 2018*). To probe the kinetics at even shorter lifetimes, SiMS of Dylight actin

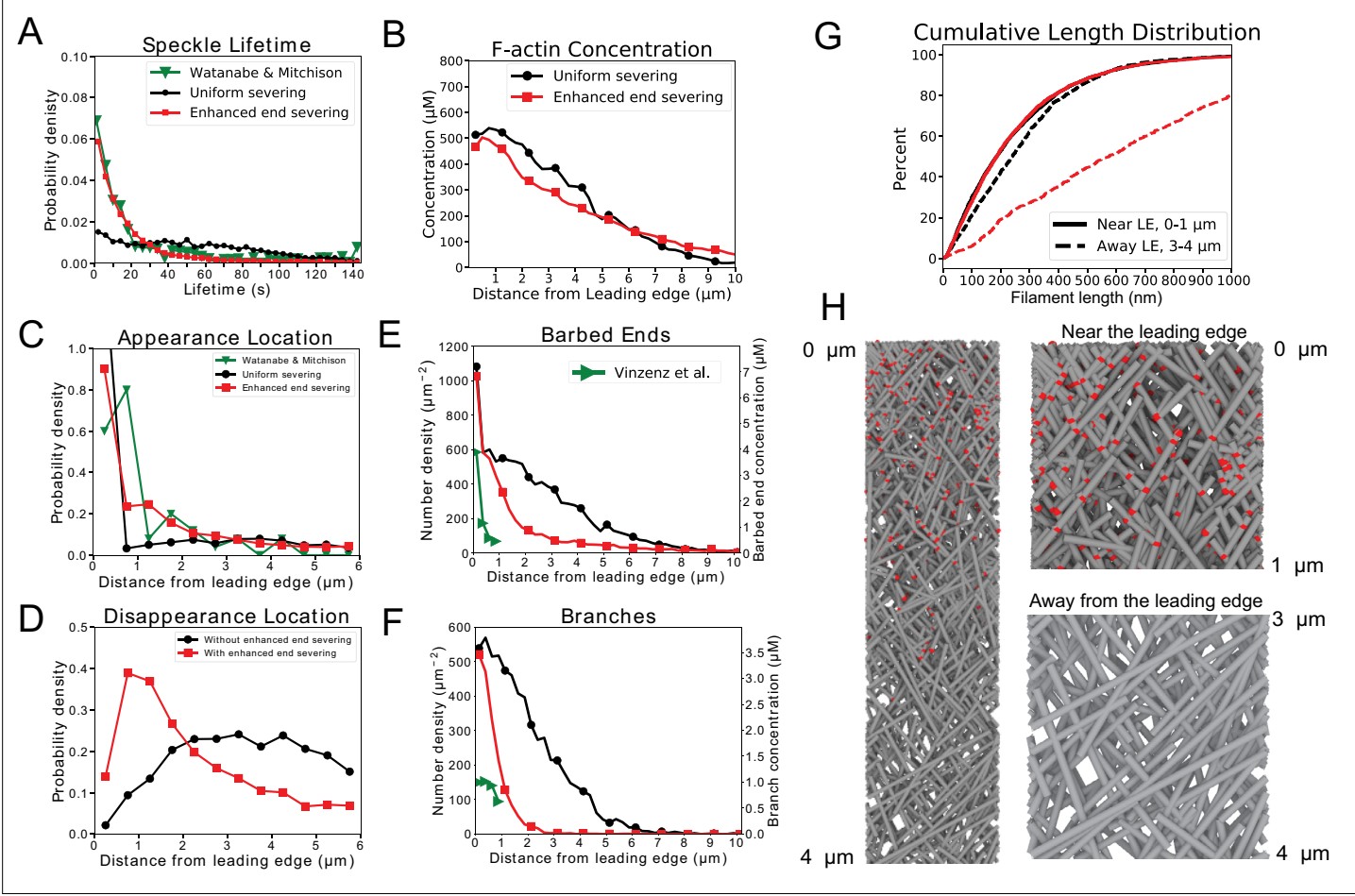

**Figure 5.** Model results for optimized parameters of model for XTC cells with uniform severing and annealing (black) or model with annealing and severing, enhanced near barbed end (red). The latter case provides good agreement with SiMS data and actin network structure; it can also lead to filament length increase away from the leading edge. (**A–H**) Panels are the same as *Figure 4* but comparing to SiMS data of *Watanabe and Mitchison, 2002* (as they were corrected for photobleaching) on XTC cells and structural data from fibroblasts by *Vinzenz et al., 2012*. In panel A, the distributions were normalized between 4 and 144 s. The fraction of oligomers is less than 0.1%. In panels E,F, the experimental data are plotted according to the left *y*-axes; the concentrations on the right y-axes use the model's assumed lamellipodial thickness. The simulation with uniform severing used $k_{unif}^{sev} = 5 \cdot 10^{-5}$ /sub/s; $l_{max}^{olig} = 80$ sub and with enhanced end severing $k_{unif}^{sev} = 5 \cdot 10^{-6}$ /sub/s; $k_{end}^{sev} = 10^{-3}$ /sub/s; $l_{max}^{olig} = 150$ sub. Other parameters listed in *Table 1* (XTC parameters).

The online version of this article includes the following figure supplement(s) for figure 5:

**Figure supplement 1.** Parameter scan of uniform and enhanced end severing rates ($k_{unif}^{sev}$, $k_{end}^{sev}$) and maximum oligomer length ($l_{max}^{olig}$), with oligomer annealing but no debranching $k_{debr} = 0$.

**Figure supplement 2.** Additional quantification of model results for XTC cells with annealing and uniform severing (top row), or annealing and severing enhanced near barbed ends (bottom row).

**Figure supplement 3.** Repeat of simulations of Figure 5 (XTC cells) with doubled branching rate $k_{br}$.

**Figure supplement 4.** Comparison of enhanced end severing simulation of *Figure 5* (XTC cells) where polymerizing ends cannot be severed to same parameter set where they can be severed.

**Figure supplement 5.** Quantification of actin turnover in simulations with severing and annealing of XTC parameters from 5 simulations in steady-state over 20 s each.

was repeated on XTC cells using the methods of *Yamashiro et al., 2014*, with a temporal resolution of 0.1 s. We measured actin speckles appearing near the leading edge, which should represent polymerization of actin monomers rather than oligomer annealing. Approximately 16% and 30% of speckles disappeared within within 0.5 and 1 s, respectively (*Figure 6A*). This large amount of short speckle lifetimes strongly supports frequent disassembly of newly polymerized F-actin near the barbed end.

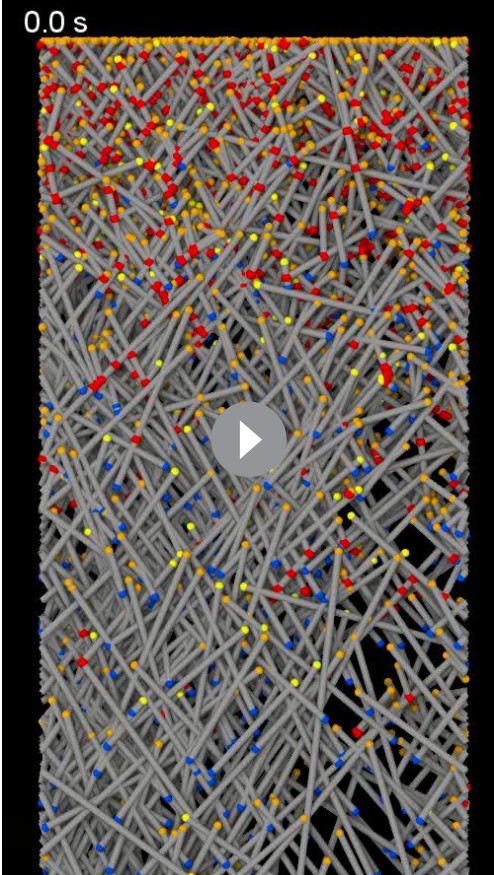

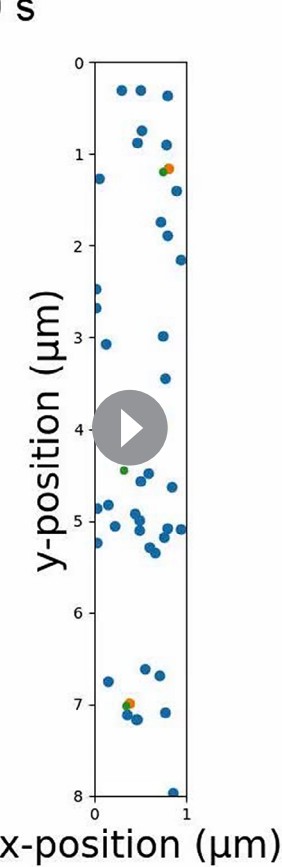

**Video 2.** Simulated XTC lamellipodium. Same as **Video 1** but using the parameters for XTC cells, enhanced end severing parameter set (Figure 5).
https://elifesciences.org/articles/69031/figures#video2

**Video 3.** Simulated keratocyte actin SiMS. Simulated actin SiMS for the enhanced end severing keratocyte parameter set (Figure 4) with 0.01% of actin monomers tracked. Speckles are positioned based on the actin monomer location. Each frame is a collection of the appearance, disappearance, and motion within 1 s. Speckles that appeared within 1 s are colored in orange and located at their appearance location. Blue speckles remained associated to the network throughout the time range and are relocating with retrograde flow. Speckles that disappeared within this time frame are colored green and located at the disappearance location. Time stamp indicates the beginning of the 1 s interval.
https://elifesciences.org/articles/69031/figures#video3

To compare to simulations, **Figure 6B** contains speckle lifetimes within 1 μm of the leading edge for the XTC parameter set with enhanced end severing and annealing of **Figure 5** (the uniform severing case of **Figure 5** did not provide as good a match to SiMS measurements and is not shown). The speckle distribution extended to short lifetimes, similar to the experiments of **Figure 6A**, with the percentage of lifetimes within 0.5 s and 1 s being 3% and 7 %, respectively. Doubling the enhanced end severing rate while also choosing parameters that reproduced speckle dynamics, lamellipodial structure and an overall increase in length ($k_{unif}^{sev} = 0$ /sub/s; $l_{max}^{olig} = 150$ sub, XTC parameters, **Table 1**), doubled the percentage of speckle short lifetimes within 0.5 s and 1 s to 7% and 14%, respectively, closer to the experimental percentages of **Figure 6A**. We conclude the short lifetimes of **Figure 6A** are within the range of what is expected by an enhanced end severing mechanism, which may even occur at a rate 2–4 times faster than the estimate of **Figure 5**.

**Kueh et al., 2008** observed that actin disassembly in lamellipodia was inhibited in the presence of cytochalasin D (CD), a barbed end capper that also inhibits binding of cofilin to F-actin at high concentrations (**Shoji et al., 2012**). We tested the effect of CD treatment on XTC cells using SiMS (**Figure 6C**). We confirmed CD's inhibitory effect on filament disassembly by observing a larger fraction of actin speckles that survive after CD treatment. These results provide further support for rapid

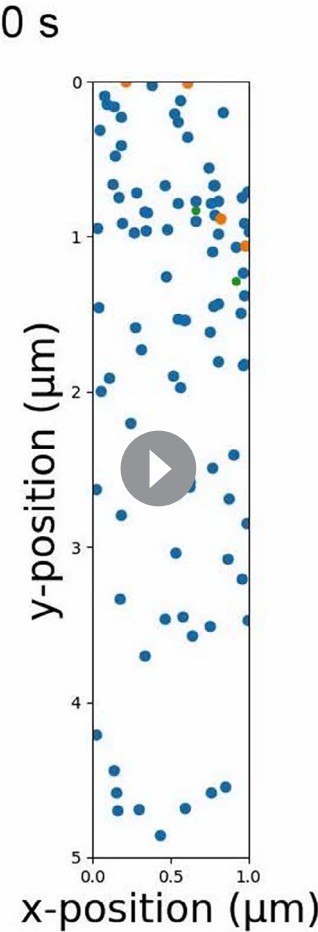

**Video 4.** Simulated XTC cell actin SiMS. Same as *Video 3* but for the enhanced end severing XTC parameter set (Figure 5) with 0.005% of actin monomers tracked.

https://elifesciences.org/articles/69031/figures#video4

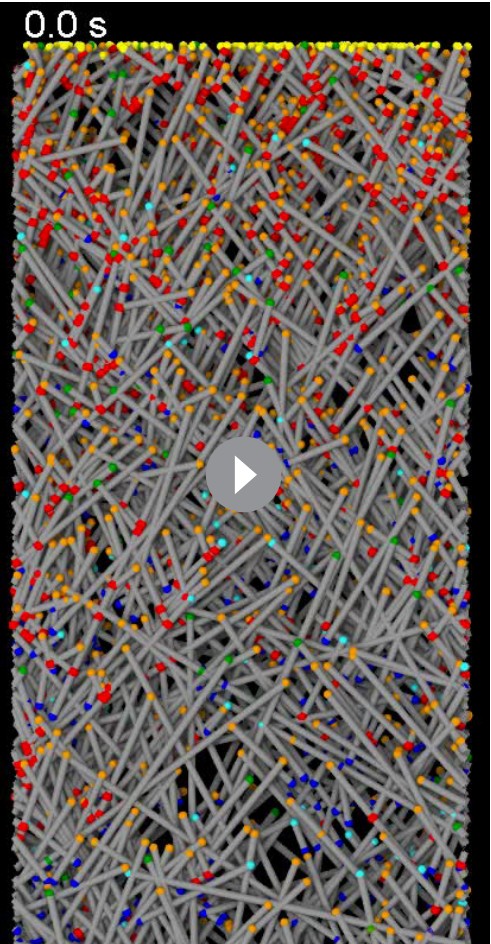

**Video 5.** Simulated keratocyte lamellipodium with frequent barbed end rapid depolymerization. Same as *Video 1*, using parameters for keratocyte cells but with no enhanced end severing, no annealing, and frequent barbed end rapid depolymerization and repolymerization. Gray lines: actin filaments; red: Arp2/3 complex; yellow:free barbed ends; orange: capped barbed ends; blue: free pointed ends not yet capped; cyan: rapidly depolymerizing barbed ends; green: repolymerizing barbed ends. Stationary green ends are barbed ends that do not polymerize because they have reached the top or bottom cell boundary (since filament kinking is turned off).

https://elifesciences.org/articles/69031/figures#video5

disassembly of newly polymerized actin at barbed ends (either through 'bursting' as proposed in *Kueh et al., 2008* or enhanced severing).

To check if tropomyosin (TPM) might be involved in the reorganization of lamellipodial networks by debranching and stabilizing progressively the longest filaments, we observed SiMS of TPM-EGFP, as described previously in *Higashida et al., 2008*, in lamellipodia of XTC cells ~30 min after cell spreading. In this early phase, the density of TPM SiMS was substantially lower in lamellipodia than in lamella (*Figure 6—figure supplement 1A*, *Video 6*). Within lamellipodia, the density of TPM SiMS becomes gradually higher toward the base of lamellipodia. TPM SiMS are scarce near the leading edge, suggesting that TPM may not have a significant effect on actin turnover near the leading edge.

We also compared the dissociation rate of tropomyosin between lamellipodia and lamella. In lamellipodia of XTC cells 30 min after cell spreading, TPM SiMS dissociated much faster in the lamellipodia than in the lamella (*Figure 6—figure supplement 1B*). TPM-EGFP was localized to actin stress fibers after cells were grown for 24 hr in the presence of serum (*Figure 6—figure supplement 1C*). The localization and kinetics of TPM SiMS are in agreement with slow assembly kinetics of TPM along

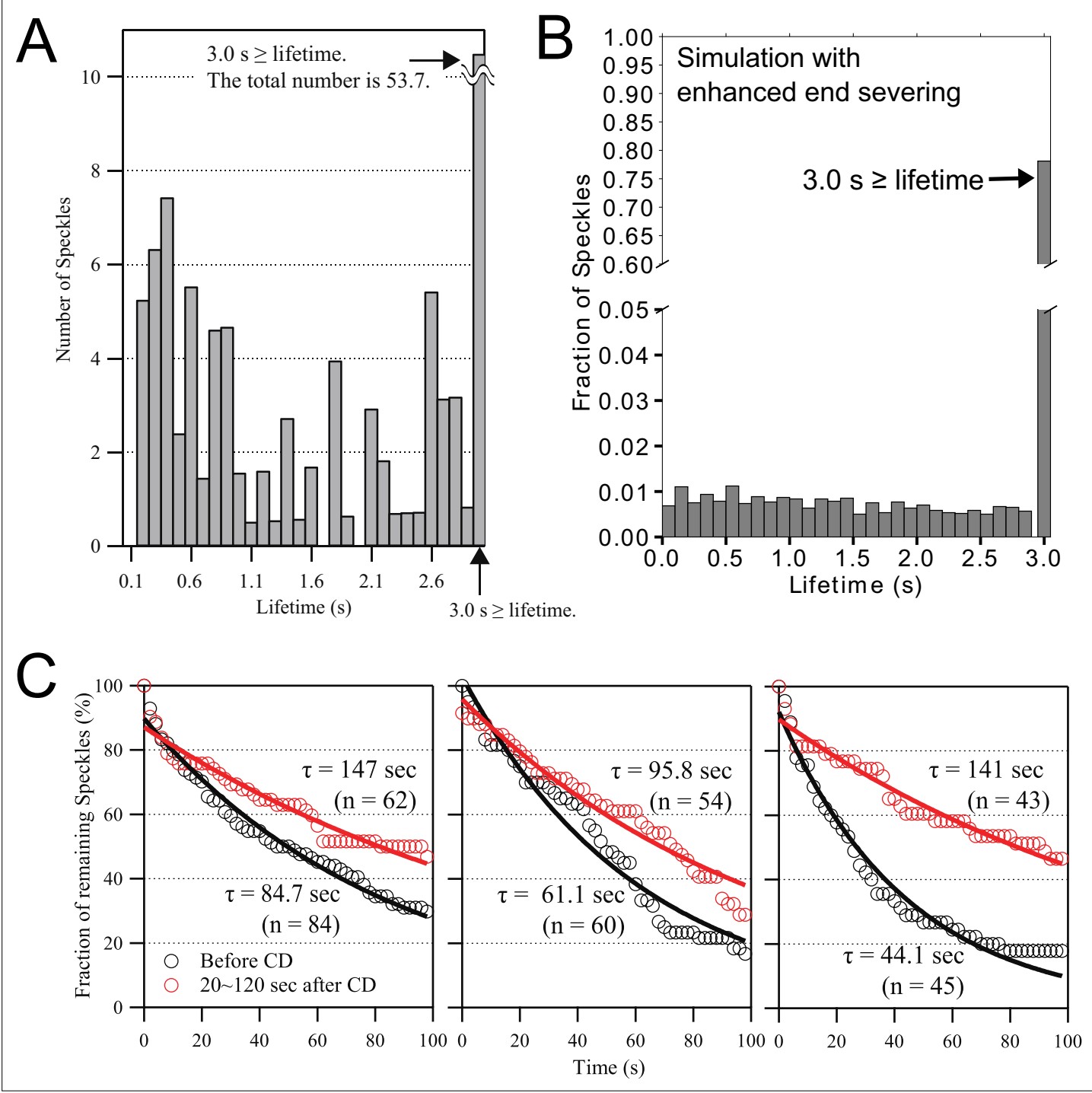

**Figure 6.** SiMS experiments support rapid disassembly of newly-assembled actin filaments near the leading edge, with disassembly inhibited by cytochalasin D. (**A**) Lifetime distribution of Dylight 550 actin speckles that appeared within ~0.5 μm of the leading edge of XTC lamellipodia, imaged for 10 s at 0.1 s/frame (n=6 cells, total number of speckles = 124). Lifetimes of 0.1 s are omitted as being beyond the temporal resolution limit. Right end bar indicates the sum of lifetime at 3 s or more. (**B**) Simulated actin speckle lifetime distribution for the case of enhanced end severing and annealing of XTC parameters in **Figure 5**, except that end severing was allowed to occur on any barbed end, including polymerizing ones (which does not have any significant influence on the results of **Figure 5**, see **Figure 5—figure supplement 4**). The fraction of lifetimes longer than 3 s were 79% (the fraction becomes 90 % if we exclude end severing of polymerizing ends). Lifetimes were averaged over 5 independent simulations and over 20 s in steady state for speckles within 1 μm of the leading edge. (**C**) The disassembly rate of Dylight 550 actin SiMS in lamellipodia of XTC cells was decreased by the treatment of 5 μM cytochalasin D (CD). The number of single-molecule speckles in lamellipodia were determined in one reference frame, and the

*Figure 6 continued on next page*

*Figure 6 continued*

reduction in the number was followed over subsequent frames. Data are from three experiments. Black and red lines are single exponential fits with decay time $\tau$. Mean of $\tau$ and the standard deviation are 63.3 ±16.6 s before treatment and 128 ±22.8 s for 20–120 s after treatment. The increase in $\tau$ is statistically significant (P=0.024; paired t-test).

The online version of this article includes the following source data and figure supplement(s) for figure 6:

**Source data 1.** SiMS SpeckleTrackerJ output for panel A.

**Source data 2.** SiMS SpeckleTrackerJ output for panel C.

**Figure supplement 1.** Tropomyosin imaging in XTC cells.

**Figure supplement 1—source data 1.** SiMS SpeckleTrackerJ output for panel B.

---

the actin filament in vitro (*Schmidt et al., 2015*). We thus conclude that at least under our observation conditions, it is unlikely that TPM stabilizes actin filaments near the tip of XTC lamellipodia.

## Discussion

We showed that the hypothesis of frequent severing and annealing (*Miyoshi and Watanabe, 2013*) provides a mechanism for distributed turnover and structural remodeling of the actin network. Using simulations based on the dendritic nucleation model, under conditions that allow self-organization into a ±35° filament orientation pattern, we determined values for uniform and enhanced end severing rates that can simultaneously account for a diverse set of experimental data: (1) actin SiMS measurements (*Watanabe and Mitchison, 2002*; *Smith et al., 2013*; *Yamashiro et al., 2018*), (2) actin photobleaching and photoactivation experiments (*Theriot and Mitchison, 1991*; *Lai et al., 2008*; *Smith et al., 2013*; *Vitriol et al., 2015*) (since our model incorporates distributed turnover *Smith et al., 2013*), (3) the presence of uncapped barbed ends through the lamellipodium (*Miyoshi et al., 2006*; *Raz-Ben Aroush et al., 2017*), and (4) the change of network structure of the lamellipodium as a function of distance to leading edge (*Svitkina and Borisy, 1999*). While there is no direct measurement of actin oligomer lengths in cells, we note that our prediction of average length of 40–80 subunits (0.11–0.21 µm, *Figure 4—figure supplement 2* and *Figure 5—figure supplement 2*) is smaller to 0.25 µm, the length of diffusing actin filaments estimated by FCS in the cortex of Hela cells (*Gowrishankar et al., 2012*) but larger than 13 subunits estimated by FRAP in keratocyte fragments (*Raz-Ben Aroush et al., 2017*).

Additionally, our simulations are consistent with SiMS of Arp2/3 complex (*Millius et al., 2012*) and capping protein (*Miyoshi et al., 2006*; *Smith et al., 2011*) in XTC cells. Agreement with SiMS of bound Arp2/3 complex lifetimes occurs because the speckle lifetimes correspond to a narrow distribution of branches as compared to the lamellipodium width (*McMillen and Vavylonis, 2016*), as in *Figure 5F*. The model is also in agreement with SiMS measurements of capping protein lifetimes in XTC cells (*Miyoshi et al., 2006*), which were used as input to the uncapping rate constant. This uncapping is important in the model, to allow uncapped barbed ends for annealing. The presence of slowly diffusing oligomers assumed in the model could also be consistent with the presence of slowly diffusing capping proteins in XTC cell lamellipodia (*Smith et al., 2011*). For the parameters of *Figures 4–5*, release of capped oligomers through end severing would

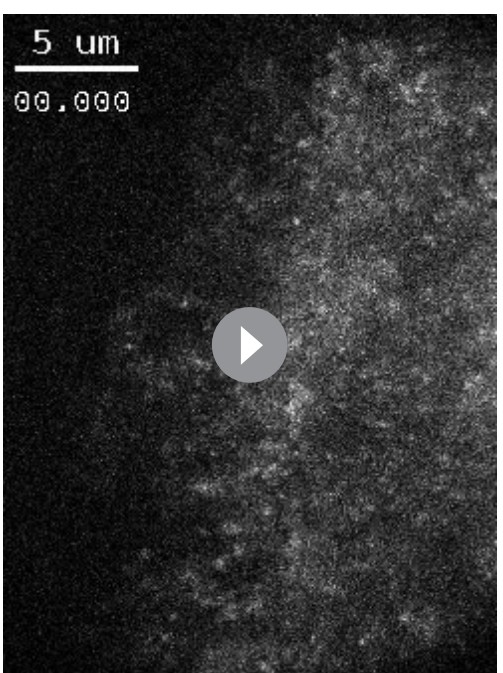

**Video 6.** SiMS of TPM-EGFP in XTC cells. The Video corresponds to Figure 6—figure supplement 1. Time shown in s.

https://elifesciences.org/articles/69031/figures#video6

not contribute significantly to capping protein SiMS lifetimes, since the corresponding rate is 4–8 times slower than uncapping.

It is also interesting to compare our model to SiMS measurement of Aip1 in XTC cells (*Tsuji et al., 2009*). Assuming that appearance of Aip1 speckles corresponds to filament disruption, the frequency of Aip1-associated filament disruption was estimated to be 1.8 µM s⁻¹ (*Tsuji et al., 2009*). We calculated the simulated overall effective severing rate by counting the total number of uniform and enhanced end severing within 20 s in steady state within 5 µm from the leading edge (average of 5 simulations). For the enhanced end severing XTC case in *Figure 5* and *Figure 5—figure supplement 3*, the effective severing rate was 0.17 µM s⁻¹ and 0.32 µM s⁻¹, respectively. These values are a few times smaller, yet not too far from the experimental estimate. This difference could indicate an even higher enhanced end severing rate in cells as compared to *Figure 5*, as also suggested by the comparison of experimental and simulated short actin lifetimes in *Figure 6A and B*.

Prior measurements of the intensity pattern of phalloidin-stained actin filaments in keratocytes treated with low doses of CD indicates shorter filaments compared to the control case (*Schaub et al., 2007*). This filament shortening could be due to capping by CD, as suggested (*Schaub et al., 2007*), however it may also be related to reduced structural remodeling of the lamellipodium through severing/destabilization near barbed ends.

Actin filament annealing, a basic assumption of our model, has been established in vitro (*Sept et al., 1999*; *Andrianantoandro et al., 2001*), including on a crowded surface which is similar to lamellipodial conditions (*Popp et al., 2007*). Our results are robust with respect to the annealing rate constant and oligomer diffusion coefficient, which is why we did not elaborate on the precise length dependence of oligomer diffusion and annealing rates. Specifically, the results are valid for a range of diffusion coefficients and annealing rate constants, as long as annealing is not dominated by reannealing to the same filament, diffusion is fast enough such that severing contributes to speckle disappearance in SiMS, and annealing does not occur further than approximately 1µm from the severing location. For example, the simulations results of *Figures 4–5* are nearly identical when reducing $k_{anneal}$ by 10 times compared to the value used from *Popp et al., 2007*. This is because both annealing rates are in a range that allows the oligomers to diffuse away from the same severed filament and large enough for annealing to remodel the network. Future work is needed however to further investigate how diffusion and annealing of filament segments occurs within the dense lamellipodial actin mesh.

Our simulations showed how severing and depolymerization regulate the length of the lamellipodial dendritic network (*Figure 2—figure supplement 1*), similar to earlier models that have been formulated at various levels of description (*Edelstein-Keshet and Ermentrout, 2001*; *Mogilner and Edelstein-Keshet, 2002*; *Carlsson, 2007*; *Michalski and Carlsson, 2010*; *Ditlev et al., 2009*; *Berro et al., 2010*; *Michalski and Carlsson, 2011*; *Lewalle et al., 2014*; *Manhart et al., 2019*), including use of explicit dendritic network (*Schreiber et al., 2010*). We also included annealing, a process previously studied using lattice models by Carlsson and Michalski (*Carlsson, 2007*; *Michalski and Carlsson, 2010*; *Michalski and Carlsson, 2011*). In their models, annealing was implemented as reappearance of lattice links and lead to a wider lamellipodium, similar to our findings. Annealing was also included in the partial differential equations model of *Ditlev et al., 2009* but the implications of this mechanism were not explicitly explored.

Closer to our work is the model by *Huber et al., 2008* who created a detailed 2D kinetic Monte Carlo model of the keratocyte lamellipodium to calculate the filament lengths and concentration profiles of actin and associated proteins. The model included diffusion of free actin monomers, filament nucleation along predefined ±35° orientations to represent branching, permanent barbed end capping, binding of ADF/cofilin and tropomyosin to filaments, as well as ATP hydrolysis and Pi release. Annealing among filaments was also included, without explicit modeling of severed filament diffusion. With this model, two distinct network regions formed (termed the lamellipodium and lamella in that paper): a region with short filaments close to the leading edge, followed by a region with longer filaments starting at about 2 µm further away.

The length increase with distance from the leading edge in the model of *Huber et al., 2008* occurred via two different mechanisms: (i) filament annealing, or (ii) polymerization of barbed ends away from the leading edge, created by severing; these ends were assumed to polymerize faster than barbed ends at the leading edge, as a result of the higher G-actin concentration away from the leading edge. The annealing mechanism (i) of *Huber et al., 2008* is different to what we described in

our work: in their study it involved the joining together of any pair of filaments at the same distance from the leading edge, without accounting of the dendritic network topology. By contrast, we assumed annealing involves diffusing oligomers. Their work also involved annealing rate constants that were one to two orders of magnitude smaller than the values in this work, and, as we understand, it was assumed that annealing occurred even with capped barbed ends. An annealing mechanism as in *Huber et al., 2008* would not contribute to actin speckle appearances and disappearances (that were not quantified in their paper) and is likely inconsistent with SiMS data. However we note that the *Huber et al., 2008* model did lead to a peak in actin disassembly at 1–2 μm away from the leading edge, similar to *Figure 4D*, as well as experimental results by qFSM in other cell types (*Ponti et al., 2004*). It's also unclear if fast polymerization away from the leading edge (mechanism (ii) in *Huber et al., 2008*) is consistent with data using FRAP (*Lai et al., 2008*; *Smith et al., 2013*) and photoactivation (*Vitriol et al., 2015*) of actin in lamellipodia of other cell types. These experiments argue against incorporation of fast-diffusing actin at the back of the lamellipodium (*Smith et al., 2013*). We also note that when we included fast polymerization of barbed ends away from the leading edge in our simulations, we typically obtained a high F-actin concentration peak away from the leading edge, unlike in the simulations of *Huber et al., 2008* or in prior experiments.

In a study of actin dynamics that combined experiment and modeling, *Raz-Ben Aroush et al., 2017* found evidence for the presence of a large pool of short actin oligomers in keratocyte lamellipodial fragments. Using FRAP on small regions, fluorescence correlation spectroscopy, and phalloidin labeling, the oligomer diffusion coefficient was estimated to be about 5 μm²s⁻¹ for oligomers with an average length of 13 subunits. *Raz-Ben Aroush et al., 2017* also report that two-thirds of actin within these fragments are diffuse, with oligomers composed a sizable fraction of this pool. A partial differential *equation 1D* model (that included polymerizable and non-polymerizable monomers, oligomers and F-actin) provided agreement with the data, assuming actin disassembly into oligomers throughout the lamellipodia and a broad distribution of polymerizing barbed ends. The finding of oligomers, as well as the proposed distributed F-actin turnover, is consistent with main assumptions of our work. However we note that the absence of local reassembly in the mechanism proposed by *Raz-Ben Aroush et al., 2017* may not easily explain experimental observations of actin FRAP or photoactivation over large regions of other cell types (*Smith et al., 2013*; *Vitriol et al., 2015*; *Yamashiro et al., 2018*), or the filament length increase across the lamellipodium. We also note that other studies (*Smith et al., 2013*; *Kiuchi et al., 2011*) have suggested much smaller concentrations of diffuse actin in lamellipodia compared to *Raz-Ben Aroush et al., 2017*, although the situation could be different in the faster keratocytes (*Yamashiro et al., 2018*) and their fragments.

The mechanism of severing and annealing modeled in this work could represent a general feature of actin dendritic networks, including yeast cells where short actin speckle lifetimes have been observed in actin patches of fission yeast (*Lacy et al., 2019*). It might provide an energetically efficient mechanism for network remodeling matching different mechanical requirements: close to the leading edge, short branched networks would provide rigidity to compressive stresses (resulting from actin polymerization against the membrane) while longer filaments at the back might be better suited for extensional stresses through myosin motors. Future work is however needed to clarify the biochemical basis of the proposed kinetics, taking into account the energetic requirements associated with ATP hydrolysis and Pi release along actin filaments, as well as mechanics and kinetics of actin filament side-binding proteins such as cofilin, GMF, and tropomyosin.

## Materials and methods

### Simulation

We developed a three-dimensional stochastic simulation of the actin network within the lamellipodium (*Figure 1*). Actin filaments are represented as straight lines without excluded volume, within the simulation box. We work in the coordinate system where the leading edge is at rest. For the purposes of this work, we do not consider explicitly the effects of ATP hydrolysis, diffusion of the actin monomer pool, or excluded volume interactions among filaments. The leading edge at y = 0, as well as the lamellipodium top and bottom at $z = 0$ and $z = 0.2 \mu m$ are hard boundaries. There is no boundary at $y \rightarrow -\infty$, allowing the mechanisms and rate constants of the system to determine the length of the lamellipodium. Periodic boundary conditions with length at least 1 μm are applied in the x-direction. A

constant relative velocity, $v_{net}$, is imposed between the network and the leading edge. The network is initialized by filament seeds of 5 subunits in length at random orientations near the leading edge. The probability of a reaction event (polymerization, depolymerization, capping, uncapping, branching, severing, and annealing) is calculated using the corresponding rates or rate constants as described below and in *Table 1*. The time step was $dt = 0.002$ s.

## Mechanisms

### Polymerization, depolymerization, capping, and uncapping

Free barbed ends, which are created by branching at the leading edge polymerize with rate $v_{pol}$. Polymerization (elongation) is simulated as stochastic increase of filament length by 2.7 nm, corresponding to addition of one monomer, when this is allowed by the distance to the hard boundaries at $z = 0$, $z = 0.2 \mu m$, or $y = 0$. We examined two different scenarios when an elongating filament reaches $z = 0$ or $z = 0.2 \mu m$: (i) the filaments stop polymerizing, or (ii) undergo 'kinking', namely they continue elongation parallel to $z = 0$ or $z = 0.2 \mu m$, at the same angle along the $xy$ plane. The latter is implemented to mimic bending of filaments when they come in contact with the membrane. Capping of free barbed ends occurs at a rate $k_{cap}$, which stops polymerization and does not allow annealing of oligomers. Capped barbed ends become free with uncapping rate, $k_{uncap}$. Free barbed ends away from the leading edge are expected to undergo different polymerization kinetics compared to barbed end at the leading edge, where membrane bound proteins such as Ena/VASP catalyze fast elongation. Recent evidence suggests that ADF/cofilin and twinfilin may assist in the depolymerization of barbed ends away from the leading edge at a slow rate (*Wioland et al., 2017*; *Johnston et al., 2015*; *Hakala et al., 2021*; *Shekhar et al., 2021*). For simplicity, and accounting for these recent observations, barbed ends formed by uncapping neither polymerize or depolymerize, still allowing however annealing of oligomers. Free pointed ends (created by severing or debranching) depolymerize with rate $v_{depol}$, implemented by stochastic length decrease by 2.7 nm corresponding to one monomer. Depolymerization stops when the filament completely depolymerizes or meets the qualifications for an oligomer. These mechanisms are depicted in *Figure 1B*.

### Branching and debranching

Filament branches are nucleated at a total rate $k_{br}$ and placed randomly along parent filaments, in proportion to their segment length within the branching region, a 27 nm region near the leading edge (*Figure 1A*), approximately the size of an Arp2/3 complex associated to proteins on the cell membrane (*Volkmann et al., 2001*). New branches form from an existing parent filament at an angle chosen from a Gaussian distribution centered at 70° with a standard deviation of 5° (*Weichsel and Schwarz, 2010*; *Gong et al., 2017*). The azimuthal angle of the branch around the axis of the parent filament is either picked from a uniform distribution (uniform 3D branching) or else uniformly but with the additional condition that the angle between the branch and the lamellipodium plane is smaller than a threshold value (typically 10°, case of branching along lamellipodium plane). Unless otherwise indicated, branching at angles larger than 80° with respect to the axis of protrusion (termed "backward" branching) is not allowed; in cases where an invalid orientation is selected, a new branch location and orientation is tried. Debranching occurs at rate $k_{debr}$, which we assume results in release of the Arp2/3 complex, leading to a free pointed end for the debranched filament. In simulations with $k_{debr} = 0$, debranching was assumed to occur for branches that become five monomers or smaller and do not contain branches of their own. Debranching also occurs when the pointed end of a parent filament depolymerizes past a branch.

### Severing

We considered uniform severing with constant rate $k_{unif}^{sev}$ per filament length, and enhanced severing near the barbed end with rate per filament length $k_{end}^{sev}$ (proposed as possible explanation for the short actin lifetimes observed by SiMS *Miyoshi et al., 2006*). Enhanced end severing occurred between the barbed end and $l_{max}^{olig}$, the longest length for an oligomer. In a severing event, a location on the filament is chosen to split the filament into two, creating a new depolymerizing pointed end and a free barbed end. An oligomer, assumed to be diffusing as discussed below, is created for any filament segment that is shorter than $l_{max}^{olig}$ and is not a branch or does not contain a branch. Oligomers formed at the barbed end retain the capped or polymerization state of the original filament. To allow a network to

form even at high severing rates, we assumed that no severing occurs within 0.1 μm of the leading edge and did not apply enhanced end severing to barbed ends polymerizing at the leading edge (unless otherwise indicated). The latter assumption does not have a significant effect on our final results (*Figure 5—figure supplement 4*).

## Oligomers and annealing

Since oligomers are short and typically anneal after a short time interval, we assumed that they do not branch, sever, or depolymerize, although we implemented capping and uncapping with the same rates as filaments. We assume that the probability of finding the end of a non-annealed oligomer displaced by distance $\delta x$ and $\delta y$ along the $x$ and $y$ directions (in the reference frame where the leading edge is at rest), after time $\delta t$, with respect to the location and time of its creation by severing, is given by 2D diffusion: $P_D(\delta x, \delta y, \delta t) = (4\pi D_{olig}\delta t)^{-1} \exp\left[-(\delta x^2 + \delta y^2)/(4D_{olig}\delta t)\right]$, where $D_{olig}$ is the oligomer diffusion coefficient. This expression neglects the small effect of advection by cytoplasmic fluid flow as well as the boundary condition at the leading edge.

Oligomer annealing to filament ends is calculated as a bimolecular reaction with rate constant $k_{anneal}$. This is implemented by scanning through all pairs of oligomers and available pointed and barbed ends, converting $P_D$ to a local oligomer concentration by assuming a uniform probability along the thin $z$ direction, and using $\delta x$, $\delta y$ as the distances between the end of the oligomer and the end on the filament that could anneal to one another. The smallest distance in the x-direction is used according to the periodic boundary conditions. If the annealing event is accepted, the length of the filament increases by the size of the oligomer. If the oligomer anneals to a barbed end, the barbed end state of the oligomer is transferred to the filament. If an oligomer did not anneal within 20 s of its creation, it was removed since in this time it likely disassembled. Removal was unlikely to occur at the chosen annealing rate constant since most oligomers annealed within a shorter time.

We checked the time step used was sufficiently small: $k_{anneal} < 120$ μM⁻¹s⁻¹ resulted in less than 40% of oligomers annealing per time step, when using the reference parameter values in *Table 1*. We also checked that the median of the time to anneal, $t_{anneal}$, and the median distance between the severing and annealing events, $r_{anneal}$, were related as expected from a theoretical approximation of these quantities assuming all barbed ends are free: $r_{anneal} = (4D_{olig}t_{anneal})^{1/2}$ with $t_{anneal} = (k_{anneal}C_B)^{-1}$, where $C_B$ is the average concentration of barbed ends in the body of the simulated lamellipodium (*Figure 4—figure supplement 3*, *Figure 5—figure supplement 5*).

## Alternative model with frequent transitions to rapid barbed end depolymerization

As an alternate mechanism to explain the short actin SiMS lifetimes, we implemented a model with frequent transitions of barbed ends to a state of rapid depolymerization with rate $v_{depol}^{cat}$. A large value of $v_{depol}^{cat}$ can mimic barbed end catastrophic disassembly (by factors such as twinfilin) or massive fragmentation (by factors such as Aip1). In the absence of annealing, this disassembly process must be matched by rapid actin repolymerization at rate $v_{repol}$ throughout the lamellipodium, to maintain the F-actin concentration in the lamellipodium.

We assume the same mechanism of polymerization, capping, uncapping, branching, and uniform severing as described in Mechanisms above, without enhanced severing near the barbed end or annealing. A new feature is the assumption that uncapping leads to either fast barbed end depolymerization or fast repolymerization, with probability $p_{cat}$ and $1 - p_{cat}$, respectively. We thus use the uncapping rate $k_{uncap}$ as a parameter to control the rate of catastrophic disassembly and the capping rate $k_{cap}$ as a rescue rate. To decouple barbed end catastrophic disassembly from severing, we now assumed that new barbed ends that form after severing are capped.

The alternate model introduces several new parameters, which are not known experimentally. However, its viability can be examined using a parameter set tuned to give similar effects to the optimized model with enhanced severing near barbed ends and annealing (*Figures 4 and 5*). Focusing on keratocytes, we keep $k_{cap}$ the same as in *Table 1* and uniform through the lamellipodium. We set $v_{depol}^{cat} = 60 \ s^{-1}$ such that the typical filament length lost in a catastrophic event, $v_{depol}^{cat}k_{cap}$, is close to the typical length of the optimized enhanced severing model, $l_{max}^{olig}/2 = 75$ in *Figure 4*. For simplicity, we set $p_{cat} = 0.5$ and $v_{repol} = v_{depol}^{cat}$ such that repolymerization balances catastrophic disassembly.

## Simulated SiMS

To compare our simulations to the actin speckle appearance and disappearance in SiMS, we tagged and tracked 1% of actin monomers that add to the network through polymerization at the leading edge. Each such polymerization corresponds to a speckle appearance. Disappearance events occur when the tagged monomers depolymerize off a pointed end or when they become part of an oligomer. Annealing of an oligomer carrying a tagged monomer is an appearance. The speckle lifetime is the time between appearance and disappearance events.

## Orientation order parameter

We counted the number of filaments having a segment within the first micrometer of the leading edge and defined the order parameter similar to *Weichsel and Schwarz, 2010* and *Mueller et al., 2017*:
$O = (N_{-70°/0°/70°} - 2N_{\pm35°})/(N_{-70°/0°/70°} + 2N_{\pm35°})$, where $N_{-70°/0°/70°}$ and $N_{\pm35°}$ are the number of filaments oriented between -20° to 20° and 60° to 80° in either direction, or between 25° - 45° in either direction, respectively. The angle is measured with respect to the axis of network growth. For 3D simulations, we used the angles of the filaments projected along the $xy$ lamellipodial plane.

## SiMS imaging experiments

SiMS experiments using Dylight 550 labeled actin introduced to XTC cells by electroporation was performed as in *Yamashiro et al., 2014*. Cells adhered on a poly-lysine-coated glass coverslip were observed by epi-fluorescence microscopy. For measurements of speckle lifetime distribution, the exposure time was 0.1 s/frame. The lifetime data were normalized for photobleaching as in *Watanabe and Mitchison, 2002*. In *Figure 6A* experiments, the leading edge was manually marked by a line which connects the centers of furthermost speckles in several consecutive images. Only actin SiMS that appeared within ~300 nm from the line were analyzed for lifetimes. Lifetimes of 0.1 s were omitted as being beyond the temporal resolution limit. Disassembly of actin speckles in lamellipodia of XTC cells were also observed after treatment with 5 μM cytochalasin D, added after several seconds from the start of observation, at 2 s intervals.

## Parameter scan for severing rates and oligomer size

To estimate parameters for uniform and enhanced end severing rates ($k_{unif}^{sev}$, $k_{end}^{sev}$), we performed a parameter search over these values as well as the maximum oligomer length $l_{max}^{olig}$. To summarize the results, we classified each set of these parameters in terms of their ability to match the actin network structure and concentration, actin speckle dynamics, and if there was an increase in length between the front and back of the lamellipodium (*Figure 3—figure supplement 1*, *Figure 4—figure supplement 1*, and *Figure 5—figure supplement 1*):

- Matching actin network structure and concentration. We marked a parameter set as satisfying this condition when the following occurred: (i) The F-actin concentration near the leading edge was 700–1500 μM for keratocytes (*Schaub et al., 2007*; *Mueller et al., 2017*) and 400–1300 μM for XTC parameters (*Watanabe, 2010*; *Vinzenz et al., 2012*); (ii) The midpoint of the concentration profile fell between 5–15 μm (keratocytes) and 3–10 μm (XTC cells); (iii) The barbed end concentration profiles were within 50% of measurements of keratocyte lamellipodia (*Mueller et al., 2017*). For XTC cells, the barbed end and branch concentrations were at least as large as those measured in fibroblast lamellipodia (*Vinzenz et al., 2012*).
- Matching actin speckle dynamics. We indicated agreement with actin speckle SiMS (*Watanabe and Mitchison, 2002*; *Yamashiro et al., 2018*; *Smith et al., 2013*) when the following conditions were met: (i) Actin speckle lifetimes peak at short times, satisfied when the probability of speckles with lifetimes between 2 and 4 s was 0.025–0.055 for keratocytes (from the full distribution ranging from 2 to 60 s) and the probability of lifetimes between 4 and 8 s for XTC cells was 0.035–0.08 (from the full distribution between 4–144 s). We did not include lifetimes shorter than 2 or 4 s, respectively, that could be at the limits of experimental resolution; (ii) The lifetime distribution extended to lifetimes longer than 10 s (at least 20%); (iii) The normalized speckle appearance and disappearance profiles as a function of distance from the leading edge were consistent with SiMS. In simulations without annealing, the speckle lifetime distribution did not peak monotonically at the shortest lifetimes, so we separately marked parameter sets with a peak within the first 15 s (keratocytes) or 35 s (XTC).

- Increase in length away from leading edge. We considered length increase to occur if the filament length at 50% of the cumulative length distribution of filaments with segments between 3–4 µm away from the leading edge was at least 0.2 µm (keratocytes) or 0.1 µm (XTC) larger compared to 0–1 µm away from the leading edge.

## Estimation of branching and capping rates by comparison to prior electron tomograms

Capping and branching rates were determined by comparing to the the barbed end, branch and filament number of electron micrographs of fibroblast cells in *Vinzenz et al., 2012* (used for the XTC parameter set) and barbed end, pointed end and filament number from *Mueller et al., 2017* for the keratocyte parameter set. In all instances of data comparison for the barbed end and pointed end number of *Mueller et al., 2017*, the average value of two 0.106 µm bins was compared to simulation results of 0.212 µm bins.

We use a simplified 2D dendritic nucleation model, without severing and annealing, to estimate the capping and branching rates for our simulations of keratocyte and XTC cases. We assume linear actin filaments, all oriented ±35° relative to the leading edge with the $-y$ direction toward the center of the cell. These filaments polymerize, branch, cap and move with respect to the position of the leading edge due to retrograde flow or cell protrusion. Filaments with free barbed ends (uncapped) remain at the leading edge and continue to polymerize. As they cap, they stop polymerization and move toward the center of the cell with retrograde flow. For the purposes of this section, we do not consider uncapping of already capped filaments. Branches form at the leading edge and also move away with retrograde flow. Denoted by $l$ the length of a filament with barbed end located at $y$, the system of equations for the number of uncapped $n_u(l,t)$, capped $n_c(l,y,t)$ barbed ends and branches $n_{br}(y,t)$ are:

$$\frac{\partial n_u(l,t)}{\partial t} = k_{br}\delta(l) - k_{cap}n_u(l,t) - v_{pol,35}\frac{\partial n_u(l,t)}{\partial l}, \qquad (1)$$

$$\frac{\partial n_c(l,y,t)}{\partial t} = k_{cap}n_u(l,t)\delta(y) - v_{net}\frac{\partial n_c(l,y,t)}{\partial y}, \qquad (2)$$

$$\frac{\partial n_{br}(y,t)}{\partial t} = k_{br}\delta(y) - v_{net}\frac{\partial n_{br}(y,t)}{\partial y}, \qquad (3)$$

The first term on the right hand side of *Equation 1* represents the branching source term which creates a new filament at 0 length. The last term of *Equation 1* is polymerization of the filaments with $v_{net} = v_{pol}\cos(35°)$. The second term of *Equation 1* and first term of *Equation 2* represents the loss of uncapped filaments and addition of capped filaments. The second term of *Equation 2* represents retrograde flow of capped filaments. *Equation 3* accounts for the generation of branches and their motion with retrograde flow toward the center of the cell.

In steady-state, the solutions of the uncapped and capped ends to be exponential and the number of branches is:

$$n_u(l) = \frac{k_{br}}{v_{pol}}e^{-l/\bar{l}} \qquad (4)$$

$$n_c(l,y) = \frac{k_{cap}k_{br}}{v_{net}v_{pol,35}}e^{-l/\bar{l}} \qquad (5)$$

$$n_{br}(y) = \frac{k_{br}}{v_{net}} \qquad (6)$$

where $\bar{l} = v_{pol,35}/k_{cap}$. To compare to electron tomogram quantifications we calculate the number of branches, barbed ends and the number of filaments in bins of increasing distance to the leading edge.

Integrating *Equation 6* over $y$ from the minimum to maximum of the bin we find the number of branches within the $i^{th}$ bin, $N_{br,i}$.

$$N_{br,i} = \frac{k_{br}\Delta y}{v_{net}}, \quad i = 1, 2, 3, \dots \qquad (7)$$

where $\Delta y$ is the bin width centered at $y = (1/2 - i)\Delta y$ such that $i = 1$ is the bin that includes the leading edge.

The number of barbed ends within a bin centered at $y$ depends on the total number of capped and uncapped filaments. We integrate over all filament lengths $l$ and the distance from the leading edge $y$:

$$N_{BE}(y) = \int_0^\infty n_u(l)\, dl \int_{y-\Delta y/2}^{y+\Delta y/2} \delta(y)\, dy + \int_0^\infty \int_{y-\Delta y/2}^{y+\Delta y/2} n_c(l,y)\, dy\, dl. \tag{8}$$

We find the solution for the number of barbed ends in the the $i^{th}$ bin to be:

$$N_{BE,i} = \frac{k_{br}}{k_{cap}} \delta_{i,1} + \frac{k_{br}}{v_{net}} \Delta y, \quad i = 1, 2, 3, \ldots \tag{9}$$

In the electron tomograms, the filament number was measured by counting the number of filaments crossing the middle plane of the measured bins. We calculate the equivalent filament number by integrating the sum of the capped and uncapped filaments that are long enough to cross the distance in $y$ at the center of such bins:

$$N_{fil}(y) = \int_{-y/\cos(35)}^\infty n_u(l)\, dl + \int_0^{-y} ds \int_{s/\cos(35)}^\infty n_c(l, y-s)\, dl. \tag{10}$$

Integrating, we find,

$$N_{fil}(y) = \frac{k_{br}}{k_{cap}} \left(2e^{\frac{yk_{cap}}{v_{net}}} - 1\right) \tag{11}$$

which can be evaluated at bin using $y = (1/2 - i)\Delta y$.

## Parameter estimation for keratocyte parameter set

*Mueller et al., 2017* quantified the number of barbed, pointed ends and filaments in keratocyte lamellipodia near the leading edge. Assuming all pointed ends near the leading edge are at a branch, we can solve *Equation 7* for the branching rate $k_{br}$ using the relative network velocity $v_{net} = 0.2\ \mu m/s$, $\Delta y = 0.212\mu m$ and an estimate for the number of branches, $N_{br}$. Unlike *Equation 7* in the simple model considered in this supplementary text, the number of pointed ends in *Mueller et al., 2017* increased over the first $0.424\mu m$ (two bins). Using the average number of pointed ends over two bins from the leading edge as an estimate of $N_{br}$ in *Equation 7* leads to $k_{br} = 152/s/\mu m$. This number is consistent with the $k_{br}$ value calculated using the number of pointed ends at longer distances, beyond the first two bins.

Using this estimate for $k_{br}$, we can estimate $k_{cap}$ in two different ways:

(i) Using *Equation 9* with $i = 1$ and *Table 2* we find the capping rate to be $k_{cap} = 1.02/s$.

(ii) Numerically solving *Equation 11*, using the number of filaments from *Table 2* with $i = 2$ and $k_{br}$ from above, we find $k_{cap} = 0.32/s$ (we do not consider $i = 1$ since the number of filaments is increasing within the region 0–0.106 µm, unlike in the current model).

**Table 2.** Barbed end, branch and filament number for fibroblast cells in *Vinzenz et al., 2012* and keratocyte cells from *Mueller et al., 2017* for lamellipodia region of 1µm.
Since the leading edge is not well defined in the EM tomograms, we consider the leading edge to begin at the maximum barbed end value but we also include the number of barbed ends that would be considered outside the cell (with this definition) in the region of 0–0.212 µm.

| Quantity | Fibroblast Region (µm) | Fibroblast Value (µm⁻¹) | Keratocyte Region (µm) | Keratocyte Value (µm⁻¹) |
|---|---|---|---|---|
| Barbed ends (first bin) | 0–0.25 | 145 | 0–0.212 | 309 |
| Barbed ends (second bin) | 0.25–0.5 | 42.5 | 0.212–0.424 | 238 |
| Branches (first bin) | 0–0.25 | 37.5 | N/A | N/A |
| Branches (second bin) | 0.25–0.5 | 37.5 | N/A | N/A |
| Pointed ends (first bin) | N/A | N/A | 0–0.212 | 91 |
| Pointed ends (second bin) | N/A | N/A | 0.212–0.424 | 231 |
| Filaments (first bin) | 0–0.25 | 150 | 0–0.106 | 200 |
| Filaments (second bin) | 0.25–0.5 | 130 | 0.106–0.212 | 256 |

In the main text and figure supplements we used branching rate and capping rates similar to the values calculated in (i) and (ii): $k_{cap} = 0.6/s$ was required to produce consistent results when depolymerization and severing was included in the simulation.

## Parameter estimation for XTC parameter set

*Vinzenz et al., 2012* studied lamellipodia of fibroblast cells and measured branch, barbed end and filament number close to the leading edge, as well as the filament length distribution *Table 2*. In this system $v_{net} = 0.03 \mu m/s$.

The branching rate can be estimated using *Equation 7* and *Table 2* to find $k_{br} = 4.5/s/\mu m$.

We can estimate the capping rate $k_{cap}$ in three different ways as follows.

(i) Using the calculation for the number of barbed ends of *Equation 9*. From *Equation 9*, the value of barbed ends in bins 2 and higher is of the same order as the number of branches in *Equation 7*, consistent with the measurements in *Vinzenz et al., 2012* in *Table 2*. The ratio of barbed ends in the first bin to the second bin is however a value that depends on $k_{cap}$ but is independent of $k_{br}$:

$$\frac{N_{BE,1}}{N_{BE,2}} = \frac{1/k_{cap} + \Delta y/v_{net}}{\Delta y/v_{net}} \tag{12}$$

Solving for the capping rate using $N_{BE,1}/N_{BE,2} = 580/170$ leads to $k_{cap} = 0.082/s$.

(ii) Using the solution for the number of filaments of *Equation 11* and the values in *Table 2*. We find for $i = 1$, $k_{cap} = 0.030/s$ and $i = 2$, $k_{cap} = 0.034/s$.

(iii) Comparison to the average filament length $\bar{l} = v_{pol,35}/k_{cap}$. The median filament branch length from *Vinzenz et al., 2012* is approximately 162 nm ≈ 60 sub (a value similar to *Bailly et al., 1999* who measured filament lengths to be between 100 and 200 nm near the leading edge of MTLn3 cells). Using this value for $\bar{l}$, we find $k_{cap} = 0.23/s$.

In the main text and figure supplements we used values of $k_{br} = 30/s/\mu m$, and $k_{cap} = 0.2/s$ to produce the same F-actin density at the leading edge while using $v_{net} = 0.05 \mu m/s$ (comparable to retrograde flow in XTC cells) and also producing consistent results when depolymerization and severing is added.

# Acknowledgements

We thank David Rutkowski for helping with network visualization. This work was supported by National Institutes of Health Grant R01GM114201 and R35GM136372. Use of the high-performance computing capabilities of the Extreme Science and Engineering Discovery Environment (XSEDE), which is supported by the National Science Foundation, project no. TG-MCB180021 is also gratefully acknowledged.

# Additional information

## Funding

| Funder | Grant reference number | Author |
|---|---|---|
| National Institutes of Health | R35GM136372 | Danielle Holz<br>Aaron R Hall<br>Eiji Usukura<br>Sawako Yamashiro<br>Naoki Watanabe<br>Dimitrios Vavylonis |
| National Institutes of Health | R01GM114201 | Danielle Holz<br>Aaron R Hall<br>Dimitrios Vavylonis |

The funders had no role in study design, data collection and interpretation, or the decision to submit the work for publication.

## Author contributions

Danielle Holz, Conceptualization, Data curation, Formal analysis, Investigation, Methodology, Software, Visualization, Writing - original draft, Writing - review and editing; Aaron R Hall, Conceptualization,

Investigation, Methodology, Software, Validation, Visualization, Writing - review and editing; Eiji Usukura, Data curation, Formal analysis, Investigation, Methodology, Writing - review and editing; Sawako Yamashiro, Conceptualization, Data curation, Investigation, Methodology, Supervision, Writing - review and editing; Naoki Watanabe, Conceptualization, Investigation, Methodology, Project administration, Resources, Supervision, Writing - review and editing; Dimitrios Vavylonis, Conceptualization, Funding acquisition, Investigation, Methodology, Project administration, Resources, Supervision, Writing - original draft, Writing - review and editing

### Author ORCIDs
Danielle Holz ⬤ http://orcid.org/0000-0002-6179-2297
Dimitrios Vavylonis ⬤ http://orcid.org/0000-0003-1802-3262

### Decision letter and Author response
Decision letter https://doi.org/10.7554/eLife.69031.sa1
Author response https://doi.org/10.7554/eLife.69031.sa2

## Additional files

### Supplementary files
• Transparent reporting form

### Data availability
All data reported in this project are present within the published figures and Supplemental Information. The code for simulations is available at https://github.com/vavylonis/LamellipodiumSeverAnneal, (copy archived at swh:1:rev:063372789ca6d92073d7db753e39b8a5051fd3d4) and will allow for all simulation plots to be reproduced. The experimental SiMS data of Figure 6 and Figure 6-supplement 1 have been provided as excel files containing the speckle tracks using the SpeckleTrackerJ ImageJ plugin.

The following dataset was generated:

| Author(s) | Year | Dataset title | Dataset URL | Database and Identifier |
|---|---|---|---|---|
| Vavylonis D | 2022 | LamellipodiumSeverAnneal | https://github.com/vavylonis/LamellipodiumSeverAnneal | GitHub, LamellipodiumSeverAnneal |

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
