## [Editor Report]

Although studied for decades, the molecular mechanisms involved in the assembly and remodeling of the lamellipodium still pose a number of questions, among which 1/ how are these networks progressively reorganized from short branched filaments to longer ones while maintaining angular order, 2/ by which mechanisms are actin filaments disassembled in these networks (depolymerization, fragmentation and/or "catastrophic" disassembly), and 3/ what is the importance and contribution of filament annealing? To address these questions, the authors develop one of the most detailed stochastic computational models to date. The model takes into account a large number of chemical reactions, including actin polymerization, depolymerization, filament branching by the Arp2/3 complex, capping, uncapping, severing, oligomer diffusion, annealing, and debranching. Close comparison of in silico and cellular actin networks allows them to evaluate the relative contribution of the different reactions. An important finding of this work is that frequent actin filament severing and annealing are phenomena that cannot be neglected to describe lamellipodial dynamics appropriately and although filament annealing in cells is not a new discovery, it is striking that it is not a negligible and inconsequential phenomenon in the cell, but contributes significantly to the reorganization of actin networks.

---

## [Decision Letter]

**Decision letter after peer review:**

Thank you for submitting your article "A mechanism with severing near barbed ends and annealing explains structure and dynamics of dendritic actin networks" for consideration by *eLife*. Your article has been reviewed by 2 peer reviewers, one of whom is a member of our Board of Reviewing Editors, and the evaluation has been overseen by Anna Akhmanova as the Senior Editor. The following individual involved in review of your submission has agreed to reveal their identity: Alex Mogilner (Reviewer #2).

Essential revisions:

(1) An important factor which could account for slower dynamics of the longer filaments present away from the leading edge is tropomyosin. Tropomyosin has been demonstrated to bind progressively to ageing branched networks, and is believed to be a key factor involved in the reorganization of these networks by debranching and stabilizing progressively the longest filaments. Would you have results or strong arguments to justify that the dynamics and reorganization of lamellipodia originate more from a severing/annealing mechanism rather than from the effect of other factors such as tropomyosin?

(2) Apart from tropomyosin, this study would be a good opportunity to discriminate between the different mechanisms of actin disassembly mentioned in the literature (beyond simple pointed end depolymerization and severing). Mechanisms such as barbed end catastrophic disassembly (by factors such as twinfilin) or massive fragmentation (by factors such as Aip1) should not be only mentioned in the discussion, but also tested as potential effective ways to disassemble actin filaments. Could you provide additional simulations, which would give arguments to validate or exclude possible actin disassembly mechanisms?

(3) Could you please provide plots of the size distribution of oligomers along the lamellipodium, which would be very informative? Does the size distribution of oligomers correspond to published values? Such distribution would also provide average oligomer sizes, which would be more informative that maximal oligomer sizes (which appear to be not so different from the average filament size). Could your model also provide, as a function of the distance from the plasma membrane, an estimate of the fraction of actin filaments generated by polymerization vs. the fraction of filaments assembled from oligomer annealing?

(4) You do not mention the density profiles of Arp2/3 and capping protein, which are known in cells and easily accessible from the simulations. Could you give these profiles and could these profiles be used to constrain even more the range of parameters?

---

## [Author Response]

Essential revisions:(1) An important factor which could account for slower dynamics of the longer filaments present away from the leading edge is tropomyosin. Tropomyosin has been demonstrated to bind progressively to ageing branched networks, and is believed to be a key factor involved in the reorganization of these networks by debranching and stabilizing progressively the longest filaments. Would you have results or strong arguments to justify that the dynamics and reorganization of lamellipodia originate more from a severing/annealing mechanism rather than from the effect of other factors such as tropomyosin?

To answer this question, we performed SiMS of tropomyosin TPM-EGFP (as described previously in Higashida et al., J. Cell Sci. 121: 3403-3412, 2008) in lamellipodia of XTC cells. TPM SiMS are scarce near the leading edge, suggesting that TPM may not have a significant effect on actin turnover near the leading edge. We also compared the dissociation rate of tropomyosin between lamellipodia and lamella and found that TPM SiMS dissociated much faster in the lamellipodia than in the lamella. We thus conclude that, at least under our observation conditions, it is unlikely that TPM stabilizes actin filaments near the tip of XTC lamellipodia.

We have thus added a new figure, Figure 6-supplement 1, Video 6, and associated discussion in the main text in the last two paragraphs before Discussion.

(2) Apart from tropomyosin, this study would be a good opportunity to discriminate between the different mechanisms of actin disassembly mentioned in the literature (beyond simple pointed end depolymerization and severing). Mechanisms such as barbed end catastrophic disassembly (by factors such as twinfilin) or massive fragmentation (by factors such as Aip1) should not be only mentioned in the discussion, but also tested as potential effective ways to disassemble actin filaments. Could you provide additional simulations, which would give arguments to validate or exclude possible actin disassembly mechanisms?

We followed the advice of the reviewers and developed a new model and performed additional simulations, the results of which are shown in new Figure 4-supplement 4 and Video 5.

The model considers a mechanism with frequent barbed end transitions to a state of rapid depolymerization. A large value of this depolymerization rate can mimic barbed end catastrophic disassembly or massive fragmentation. This process must be matched by rapid actin re-polymerization, to maintain the F-actin concentration through the lamellipodium. We show that such a model can come rather close to matching model results to the three main experimental criteria considered: agreement with SiMS data, increase in filament length with distance from the leading edge, broad F-actin concentration profile.

However, as we mention in the main text, such a mechanism is unlikely to explain prior FRAP or photoactivation (PA) data of GFP-actin, such as the slow FRAP recovery at the back of the lamellipodium. The severing and annealing mechanism, by contrast, can be consistent with FRAP/PA studies as a result of the local reassembly of slowly diffusing oligomers. We hope these additional simulations as well as the associated discussion we provide will be beneficial to the field.

(3) Could you please provide plots of the size distribution of oligomers along the lamellipodium, which would be very informative? Does the size distribution of oligomers correspond to published values? Such distribution would also provide average oligomer sizes, which would be more informative that maximal oligomer sizes (which appear to be not so different from the average filament size). Could your model also provide, as a function of the distance from the plasma membrane, an estimate of the fraction of actin filaments generated by polymerization vs. the fraction of filaments assembled from oligomer annealing?

We added plots of the distribution of oligomer sizes in Figure 4-supplement 2 and Figure 5-supplement 2, for the keratocyte and XTC cases, respectively. The approximately-uniform shape of the oligomer size distribution reflects our assumption of uniform severing rate across the filament end region. The distribution is also influenced by oligomer formation by depolymerization, debranching and severing resulting into two oligomers. The average oligomer size is approximately half of the maximum oligomer size parameter in our model and is in the range of 40-80 subunits, or 0.1-0.2 *µ*m.

We are not aware of any direct experimental measurement of oligomer size in cells. We added a sentence in the first paragraph of Discussion noting that measurements in the cortex of Hela cells by Gowrishankar et al., indicated diffusing F-actin filaments of average length 0.25 *µ*m. We also note that Aroush et al., estimate oligomer lengths of 13 subunits in keratocyte fragments using FRAP. The estimate by Aroush et al., relied on the assumption of a large concentration of non-annealing oligomers, which is different to our proposed mechanism, as we had discussed in the Discussion section.

We now provide graphs showing the origin of actin filaments through polymerization or annealing in Figure 4-supplement 2 and Figure 5-supplement 2, for the keratocyte and XTC cases, respectively. These graphs show that F-actin changes from being primarily monomeric in origin at the front to primarily oligomeric at the back.

(4) You do not mention the density profiles of Arp2/3 and capping protein, which are known in cells and easily accessible from the simulations. Could you give these profiles and could these profiles be used to constrain even more the range of parameters?

We did not add separate figures for the density profiles of Arp2/3 complex since these are the same as plots of the concentration of branches in Figures 4 and 5. The distribution of Arp2/3 complex (branches) in Figure 4 and 5 is narrower compared to F-actin and comparable in size to experimental values. We added clarifying comments when presenting these panels in the main text. We had already taken the Arp2/3 complex profiles into account by assuming a rate of spontaneous debranching, or debranching of short branches after severing. Thus, these Arp2/3 complex profiles do not provide additional constraints beyond what we had already considered. We added a note about this in Table 1.

In the new Figure 4-supplement 2 and Figure 5-supplement 2, we now show the fraction of capped ends as a function of distance from the leading edge. For annealing to have an effect, a significant number of barbed ends must be available for annealing, so any capping mechanism (through capping protein specifically or other protein) should be transient. The provided capped concentration profiles do show a significant fraction of uncapped ends, though the precise value of this ratio or its profile across the leading edge is not heavily constrained by our model. Thus in this study we did not try to fit capping protein profiles (which have been found to vary across lamellipodia of different cell types, e.g. by Miyoshi et al., 2006, Iwasa and Mullins 2007, Lai et al., 2008)